# Sterile inflammation via *TRPM8* RNA-dependent TLR3-NF-kB/IRF3 activation promotes antitumor immunity in prostate cancer

Alessandro Alaimo [1,17 ✉], Sacha Genovesi [1], Nicole Annesi[1], Dario De Felice [1], Saurav Subedi[2], Alice Macchia[1], Federico La Manna[2], Yari Ciani[1], Federico Vannuccini [1], Vera Mugoni[1], Michela Notarangelo [1], Michela Libergoli [1], Francesca Broso [1], Riccardo Taulli[3,4], Ugo Ala [5], Aurora Savino[6], Martina Cortese [1], Somayeh Mirzaaghaei [6,7], Valeria Poli [6,7], Ian Marc Bonapace[8], Mauro Giulio Papotti[9], Luca Molinaro[9], Claudio Doglioni[10], Orazio Caffo[11], Adriano Anesi[12], Michael Nagler[13], Giovanni Bertalot [14,15], Francesco Giuseppe Carbone[14], Mattia Barbareschi [14,15], Umberto Basso[16], Erik Dassi[1], Massimo Pizzato[1], Alessandro Romanel [1], Francesca Demichelis[1], Marianna Kruithof-de Julio[2,13] & Andrea Lunardi [1,17 ✉]

## Abstract

Inflammation is a common condition of prostate tissue, whose impact on carcinogenesis is highly debated. Microbial colonization is a well-documented cause of a small percentage of prostatitis cases, but it remains unclear what underlies the majority of sterile inflammation reported. Here, androgen- independent fluctuations of PSA expression in prostate cells have lead us to identify a prominent function of the *Transient Receptor Potential Cation Channel Subfamily M Member 8 (TRPM8)* gene in sterile inflammation. Prostate cells secret *TRPM8* RNA into extracellular vesicles (EVs), which primes TLR3/NF-kB-mediated inflammatory signaling after EV endocytosis by epithelial cancer cells. Furthermore, prostate cancer xenografts expressing a translation-defective form of *TRPM8* RNA contain less collagen type I in the extracellular matrix, significantly more infiltrating NK cells, and larger necrotic areas as compared to control xenografts. These findings imply sustained, androgen-independent expression of *TRPM8* constitutes as a promoter of anticancer innate immunity, which may constitute a clinically relevant condition affecting prostate cancer prognosis.

**Keywords** Prostate; Inflammation; TRP; PSA; Immunity
**Subject Categories** Cancer; Immunology; Membranes & Trafficking

## Introduction

The term inflammation describes a condition where eukaryotic cells react to a wide range of dangerous stimuli by releasing specific molecular signals aimed at activating the immune response. The presence of pathogens is the main cause of an inflammatory state. Epithelial cells, fibroblasts, and immune cells express proteins collectively defined Pattern Recognition Receptors (PRRs) with the precise function of intercepting microbic molecules (DNA, RNA, proteins, lipids; pathogen-associated molecular patterns, PAMPs) both inside and outside the cell. Well characterized PRRs are the Toll-like receptors (TLRs), cytoplasmic NOD-like receptors (NLRs), intracellular retinoic acid-inducible gene I-like receptors (RLRs), transmembrane C-type lectin receptors (CLRs), absent in melanoma 2-like receptors (AIM2s) and cGAS/STING. Once engaged, PRRs activate a signal cascade reaching effector proteins such as STATs and NF-kB whose primary goal is to induce the transcription of pro-inflammatory cytokines and chemokines that are then secreted by the cell (Cao, 2016; Li and Wu, 2021).

[1]Department of Cellular, Computational and Integrative Biology (CIBIO), University of Trento, Trento, Italy. [2]Department for BioMedical Research, Urology Research Laboratory, University of Bern, Bern, Switzerland. [3]Department of Oncology, University of Torino, Torino, Italy. [4]Center for Experimental Research and Medical Studies (CeRMS), AOU Città della Salute e della Scienza di Torino, Torino, Italy. [5]Department of Veterinary Sciences, University of Torino, Torino, Italy. [6]Department of Molecular Biotechnology and Health Sciences, University of Torino, Torino, Italy. [7]Molecular Biotechnology Center (MBC) "Guido Tarone", University of Torino, Torino, Italy. [8]Department of Biotechnology and Life Sciences, University of Insubria, Busto Arsizio, VA, Italy. [9]Department of Pathology, University of Torino and AOU Città della Salute e della Scienza di Torino, Torino, Italy. [10]Division of Pathology, Pancreas Translational and Clinical Research Center, San Raffaele Scientific Institute IRCCS Vita Salute, San Raffaele University, Milano, Italy. [11]Medical Oncology Department, Santa Chiara Hospital-APSS, Trento, Italy. [12]Operative Unit of Clinical Pathology, Santa Chiara Hospital-APSS, Trento, Italy. [13]Department of Urology, Inselspital, Bern University Hospital, University of Bern, Bern, Switzerland. [14]Operative Unit of Anatomy Pathology, Santa Chiara Hospital-APSS, Trento, Italy. [15]Centre for Medical Sciences-CISMed, University of Trento, Trento, Italy. [16]Oncology 1 Unit, Department of Oncology, Istituto Oncologico Veneto IOV IRCCS, Padova, Italy. [17]These authors contributed equally: Alessandro Alaimo, Andrea Lunardi. ✉E-mail: alessandro.alaimo@unitn.it; andrea.lunardi@unitn.it

A second type of inflammation is the so-called *sterile inflammation*. In this case, the triggers are molecules released by dying cells due to trauma, ischemia, tissue aging, or by cancer cells (damage-associated molecular patterns, DAMPs). Both types of inflammation are characterized by the activation of multiple PRRs at the very same time due to the concomitance of different PAMPs or DAMPs that channel a large volume of downstream signals to the effector proteins. As an integral part of innate and adaptive immunity, the acute inflammatory response is a crucial safeguard mechanism, regardless of the underlying trigger (Chen and Nuñez, 2010).

Much more controversial are the origins of chronic inflammation and the consequences of this condition on tissues and organs health and function. Cirrhosis and Inflammatory Bowel Disease are classical examples of chronic inflammatory disorders with proven causal roles in liver and colorectal tumorigenesis, respectively (Axelrad et al, 2016; Keller et al, 2019; Pinter et al, 2016). This link is much more subtle for other tissues due to a growing body of conflicting clinical and preclinical evidence. Prostatitis, benign prostatic hyperplasia (BPH) and prostate cancer (PCa) are common diseases associated with aging in men (Ørsted and Bojesen, 2013). Inflammation is a typical condition across the spectrum of prostate pathologies and one of the most debated topics in prostate disease. Considered a major player in BPH and prostate carcinogenesis according to pre-clinical models (de Bono et al, 2020), clinical studies associate chronic inflammation of the prostate gland with a lower tumor grade and a better prognosis compared to malignancies developed in noninflamed organ (Naha et al, 2021; Zhang et al, 2019). Infections are responsible for only 10% of cases, while obesity, diet and tobacco smoking are all factors commonly indicated as possible causes of chronic inflammation. Overall, the origin of sterile inflammatory states in tissues and organs is very often elusive, and the underlying molecular mechanisms are still controversial (Ørsted and Bojesen, 2013). Stratifying the different types of inflammation based on molecular drivers, sensors, effectors and magnitude of response can help untangle ambiguities about the effect the inflammatory process may cause in the specific context under examination.

Here, we demonstrate that the RNA of *Transient Receptor Potential Cation Channel Subfamily M Member 8* (*TRPM8*), a gene highly expressed in the prostate epithelium, triggers a sterile inflammatory condition in both normal and tumoral prostate cells via Toll-like receptor 3 (TLR3). In vitro, sterile inflammation promotes the AR-independent expression of *KLK3* gene coding for PSA but leaves cell proliferation and survival unaffected. In vivo, when transplanted in BALB/c nude mice, LNCaP prostate cancer cells experiencing TRPM8 RNA-induced sterile inflammation form tumors characterized by low collagen I deposition, high numbers of infiltrating Natural Killer (NK) cells, and widespread necrosis.

This study deciphers a novel molecular mechanism of aseptic inflammation in the prostate epithelium caused by the interaction of TRPM8 RNA with TLR3, which exerts a tumor-suppressive role in the prostate by promoting the safeguarding activity of innate immunity.

# Results

## TRPM8 regulates PSA levels in prostate cells regardless of the androgen pathway

*TRPM8* encodes for a 128 kDa cation channel. Based on previous literature and our recent work, *TRPM8* expression characterizes the luminal compartment of the prostate epithelium in mammals, showing different levels of correlation with the expression of classical androgen target genes such as *TMPRSS2*, *KLK2*, *NKX3.1*, and *KLK3* both in normal tissue and in hormone naive prostate cancer (PCa) (Alaimo et al, 2020; Genovesi et al, 2022; Lunardi et al, 2021). Moreover, androgens and PSA have been proposed as endogenous agonists of the channel (Asuthkar et al, 2015; Gkika et al, 2010), thus suggesting a possible operative network linking TRPM8 function to AR activity.

In LNCaP hormone-sensitive metastatic prostate cancer cells genetically modified to express different amounts of TRPM8 (Alaimo et al, 2020), *TRPM8* expression paralleled *KLK3* (PSA) expression at both RNA and protein levels, while leaving unchanged other canonical AR transcriptional targets, such as *KLK2*, *NKX3.1* and *TMPRSS2* (Figs. 1A and EV1A). Transient knock-down of *TRPM8* (Fig. 1B,C) and the correlation between *TRPM8* and *KLK3* gene expression in human samples from the TCGA database (Fig. EV1B) strengthened this finding. Unexpectedly, the results were recapitulated even under complete androgen blockade (Fig. 1D). To further explore the connection between *TRPM8* and *KLK3* genes, we expanded our analysis on the immortalized prostate cell line RWPE-1, which is characterized by a very low activity of the androgen pathway and undetectable PSA protein. RWPE-1 cells stably expressing exogenous *TRPM8* RNA (Alaimo et al, 2020) showed a significant increase of *KLK3* mRNA and PSA levels (Figs. 1E–G and EV1C,D), which was unaffected by complete androgen blockade (Fig. EV1E). Along with the evidence of an AR-independent mechanism, the PSA amount in both LNCaP and RWPE-1 cells was not influenced by gating TRPM8 channel with its potent agonist WS-12 or antagonist AMBT (Fig. 1H,I), suggesting that channel activity is not involved in the observed effects associated with TRPM8 overexpression.

## Extracellular vesicles secreted TRPM8 RNA induces Toll-like receptor 3 (TLR3) signaling

In recent years, a growing amount of works has changed the field of biology describing a plethora of molecular functions played by long non-coding RNAs (Mattick et al, 2023). To untangle possible translational-independent functions of *TRPM8* RNA that could help explain the correlation between *KLK3* mRNA and PSA levels, we mutagenized the TRPM8 cDNA to replace methionine +1 and +10 to leucine (Fig. 2A). TRPM8 mutant transcript (MM) was stable (Figs. 2B and EV2A), roughly 50 times more abundant than endogenous TRPM8 RNA (Fig. 2C), but unable to encode for the protein in both LNCaP and RWPE-1 TRPM8 knock-out (CAS) cell lines and in PC3 cells (Fig. EV2B,C). Nevertheless, *TRPM8* MM RNA expression raised *KLK3* RNA level (Fig. 2D and EV2D,E) and PSA amount (Fig. 2E,F) in both cell lines.

Interestingly, NF-kB has been demonstrated to promote AR-independent *KLK3* gene transcription in prostate cells (Chen and Sawyers, 2002). Cell fractionation showed an increase in the amount of NF-kB p65 in the nucleus of LNCaP cells expressing endogenous levels of *TRPM8* RNA (LNCaP WT) compared with LNCaP cells in which *TRPM8* expression was eliminated (LNCaP CAS) (Figs. 2G and EV2F). In contrast, NF-kB p65 localized exclusively in the cytosol of RWPE-1 WT and RWPE-1 CAS cells, consistent with the very low amount of TRPM8 RNA in these cells (Fig. 2H). NF-kB p65 mainly localized in the nucleus of RWPE-1

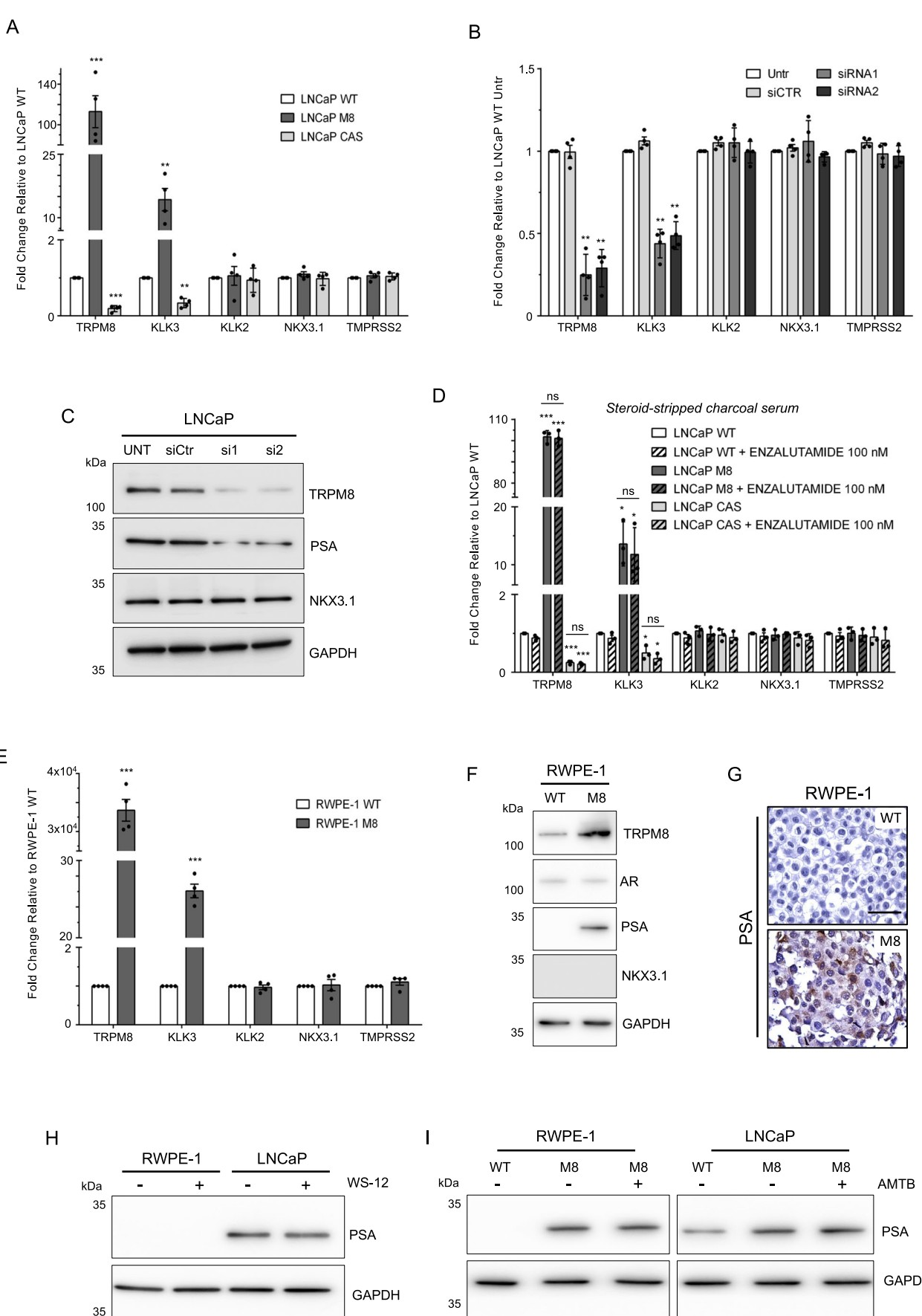

**Figure 1.  TRPM8 affects PSA levels independently from AR activity in prostate cells.**

(A) RT-qPCR analysis of *TRPM8*, and AR transcriptional targeted genes *KLK3, KLK2, NKX3.1, TMPRSS2* in wild type (WT) and genetically modified (*TRPM8* overexpression, M8; *TRPM8* knock-out, Crisper/CAS9 hereafter CAS) LNCaP cells. (B,C) RT-qPCR (B) and Western blotting (C) of LNCaP cells transfected with non-targeting (siCTR) or *TRPM8* targeting (siRNA1 and siRNA2) small interfering RNA molecules. (D) RT-qPCR analysis of wild type (WT) and genetically engineered (M8 and CAS) LNCaP cells cultured in steroid-stripped charcoal serum and treated with 100 nM Enzalutamide for 48 h. (E,F) RT-qPCR (E) and immunoblot (F) analyses showing expression of *TRPM8, KLK3* (PSA) and the indicated genes in RWPE-1 wild type (WT) and RWPE-1 stably overexpressing *TRPM8* (M8). (G) Immunohistochemistry of PSA on paraffin embedded RWPE-1 WT and RWPE-1 M8 cell pellets. Scale bar, 50 µm. (H,I) Immunoblotting analysis showing expression of PSA in RWPE-1 and LNCaP cells treated or not with WS-12 (TRPM8 agonist, 1 µM, 12 h, (H)) and treated or not with AMTB (TRPM8 antagonist, 10 µM, 24 h, (I)). GAPDH was used as loading control. Data Information: In (A,B and D,E) data are presented as mean ± SD of n = 4 (A,B,E) and n = 3 (D) independent biological replicates. *P ≤ 0.05; **P ≤ 0.01; ***P ≤ 0.001; ns, not statistically significant (Two-tailed Student's *t*-test). Source data are available online for this figure.

and LNCaP prostate cells when the coding (M8) or noncoding (MM) transcript of TRPM8 was exogenously expressed in these cells (Figs. 2G–I and EV2F), along with the canonical NF-kB and IRF3 targeted genes that were markedly induced (Fig. EV2G,H). Notably, the morphology, proliferation and survival rates, and migratory and invasive capabilities of prostate cells remained unchanged (Appendix Fig. S1; and ref. (Alaimo et al, 2020)). In support of a molecular circuit connecting *TRPM8* transcript to innate immunity (Fig. 2L), the *TRPM8* transcript was recently described in the blood of PCa patients (De Souza et al, 2020). Specifically, we detected *TRPM8* RNA into extracellular vesicles (EVs) collected from the supernatant of LNCaP wild type (WT) and MM cell lines (Figs. 3A and EV3A), as well as into EVs isolated from the blood of PCa patients (Fig. 3B). To investigate if the secretion of *TRPM8* RNA was responsible for NF-kB activation and PSA induction (Fig. 3C), LNCaP and RWPE-1 CAS cell lines were conditioned with the supernatants of WT (LNCaP only), CAS, M8, and MM LNCaP and RWPE-1 cell lines, respectively. Consistently, supernatants of WT, M8, and MM cells triggered NF-kB nuclear shuttling and increased PSA levels in *TRPM8* knocked-out (CAS) LNCaP and RWPE-1 cells (Figs. 3D and EV3B,C). Isolation of EVs from supernatants of CAS, WT, and MM LNCaP cells formally proved the direct involvements of EVs-delivered *TRPM8* RNA in NF-kB activation in prostate cells (Fig. 3E). Pattern-recognition receptors (PRR) are a special class of proteins that function as sensors for microbial molecules in eukaryotic cells (Wicherska-Pawłowska et al, 2021). Among them, the Toll-like family of receptors are transmembrane proteins localized both in the plasma membrane (TLR-1, TLR-2, TLR-4, TLR-5, TLR-6) and endosomes (TLR-3, TLR-7, TLR-9), with the latter recognizing endocytosed single- and double-stranded RNAs (Wicherska-Pawłowska et al, 2021; Medzhitov, 2001). Once activated by binding to their ligands, TLRs signaling triggers the nuclear shuttling of NF-kB and the transcriptional induction of a potent pro-inflammatory program (Karin, 2006; Kawai and Akira, 2007). Endosomal TLRs activate NF-kB via TGF-beta Activated Kinase 1 (TAK1), which promotes nuclear translocation and transcriptional activity of p65 through phosphorylation of serine 536. Further strengthening the hypothesis of a spurious endosomal TLR signaling, treatments with the TAK1 inhibitor Takinib abolished NF-kB activation and PSA induction in both RWPE-1 and LNCaP cell lines expressing either coding (M8) or non-coding (MM) forms of the *TRPM8* transcript (Fig. EV3D,E). Of the three TLRs (TLR3, TLR7 and TLR8) that operate in the endosomal compartment as RNA sensors, TLR3 is the most represented in RWPE-1 and LNCaP cell lines (Fig. EV4A). TLR3 is a sensor of viral infection and sterile tissue necrosis through the recognition of double-stranded RNAs (Tatematsu et al,

2013). In silico prediction of TRPM8 RNA structure showed several highly stable stems with bulge/internal loops that are considered potent TLR3 agonists (Fig. EV4B,C). Pharmacological inhibition of TLR3 (Figs. 3F,G and EV4D,E), but not TLR7/8, RIG1, and PKR (Figs. 3G and EV4F), turned off the signal in both LNCaP and RWPE-1 cell lines thus demonstrating the involvement of this receptor in *TRPM8* RNA sensing and activation of an extensive cascade of inflammatory molecular circuits (Fig. EV4E). Finally, both native RIP and fCLIP experiments formally proved the binding of different fragments of the endogenous (WT) and exogenous (MM) *TRPM8* RNA to the immunoprecipitated TLR3 RNA sensor (Figs. 3H–J and EV4G, Appendix Fig. S2).

## TRPM8-induced sterile inflammation shapes prostate tumor microenvironment

While on one side acute inflammatory response is a crucial safeguard mechanism, on the other side chronic inflammation can lead to serious clinical conditions. Cirrhosis, a chronic inflammatory status of the liver, is associated with the increased incidence of hepatocellular carcinoma (HCC) (Pinter et al, 2016), while inflammatory bowel diseases (IBD) such as ulcerative colitis or Crohn's disease increase the risk of developing colorectal cancer (CRC) (Axelrad et al, 2016; Keller et al, 2019). Remarkably, the robust pro-inflammatory signature elicited in both LNCaP and RWPE-1 cell lines by *TRPM8*-dependent NF-kB activation did not interfere with their features *in vitro* (Appendix Fig. S1). In vivo, WT and MM LNCaP cells injected subcutaneously in BALB/c *nude* mice showed comparable tumorigenic potential at a macroscopic level (Fig. 4A,B). Nevertheless, histopathological analyses revealed several important features discriminating WT from MM xenografts. Western blot and immunohistochemistry confirmed a higher amount of PSA and a greater percentage of PSA-positive LNCaP cells in MM compared with WT xenografts (Figs. 4C and EV5A,B), while quantification of circulating total (free + complexed) PSA in mouse plasma suggested the opposite (Fig. EV5C). Trichrome staining highlighted lower deposition of collagen type I in xenografts generated by LNCaP MM cells compared to those obtained with WT LNCaP (Figs. 4D and EV5D), a condition that is further proved by Western blot analysis of Col1A1 (Fig. 4C). Cancer Associated Fibroblasts are the main producers of collagens in the cancer microenvironment. Analysis of fibroblasts markers α-Smooth muscle actin (α-Sma) and Vimentin showed a significant increase of the latter in fibroblasts surrounding the LNCaP MM cells (Figs. 4C,E and EV5E). To shed some light on the lower deposition of collagen type I, we analyzed COL1A1 production by human fibroblasts isolated from surgically resected normal and prostate cancer specimens (Fig. EV5F). Normal Associated Fibroblasts (NAFs) and Cancer Associated

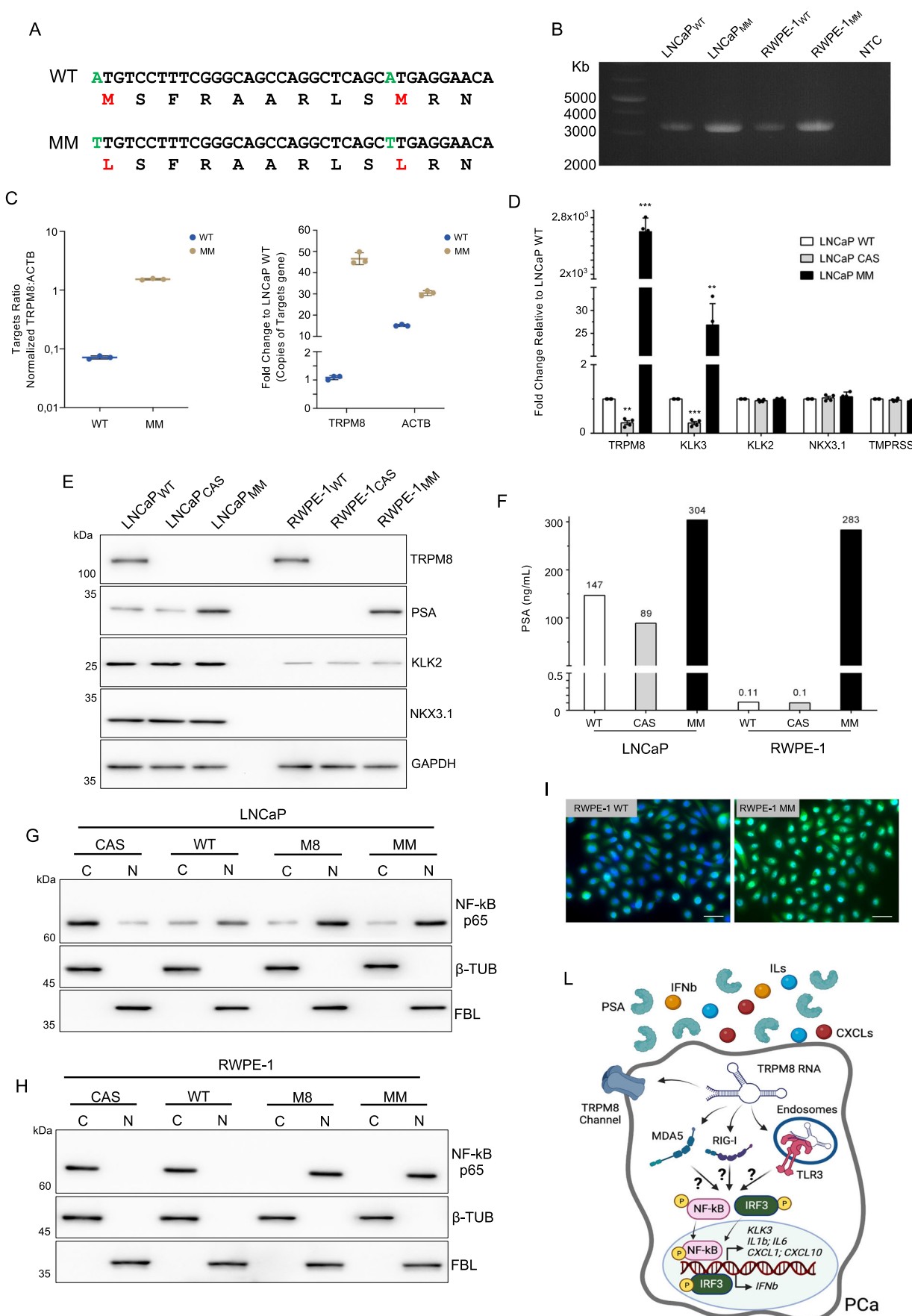

**Figure 2. TRPM8 RNA increases KLK3 RNA and PSA amount.**

(A) Sequences alignment of wild type (WT) and translationally defective mutant (MM) TRPM8. (B) End-point PCR analysis showing the stability of full-length TRPM8 MM RNA in both LNCaP and RWPE-1. Expression of endogenous TRPM8 RNA (WT) was used as reference. (C) ddPCR quantification of TRPM8 gene expression in WT and MM. Concentration of TRPM8 and ACTB. (left panel) and total number of TRPM8 copies normalized to total number of ACTB copies (right panel). (D) RT-qPCR analysis of *TRPM8*, *KLK3* and other known AR-targeted genes in both wild type (WT) and genetically engineered (CAS and MM) LNCaP cell lines. (E) Western blotting analysis of TRPM8, PSA, KLK2, and NKX3.1 in both wild type (WT) and genetically engineered (CAS and MM) LNCaP and RWPE-1 cell lines. GAPDH was used as loading control. (F) PSA amount (ng/ml) in the indicated cell-conditioned culture medium. (G,H) Immunoblot analysis of NF-kB p65 in cytosolic and nuclear fractions of CAS, WT, M8, and MM engineered LNCaP (G) and RWPE-1 (H) cell lines. β-Tubulin (β-TUB) and Fibrillarin (FBL) were used as cytoplasmic and nuclear markers, respectively. (I) Immunolocalization of NF-kB in wild type (WT) and MM RWPE-1cells. Scale bare 10 μm. (L) Schematic representation of TRPM8 RNA mediated activation of NF-kB/IRF3 response (*Created with BioRender.com*). Data Information: In (C,D) data are presented as mean ± SD of $n = 3$ (C) and $n = 4$ (D) independent biological replicates. In (F) data are presented as results of a single experiment. **$P \leq 0.01$; ***$P \leq 0.001$ (Two-tailed Student's *t*-test). Source data are available online for this figure.

Fibroblasts (CAFs) were either maintained in their medium (CTR) or conditioned with the supernatants of LNCaP WT, CAS, and MM. Western blot analysis showed a substantial reduction of COL1A1 protein only in CAFs exposed to the MM supernatant (Fig. 4F). The TCGA datasets underscored a mild but significant anticorrelation between *TRPM8* and *P3H3* and *COLGALT1*, two essential genes of the metabolism of type I collagen (Fig. EV5G). Both genes resulted significantly downregulated in CAFs conditioned with the supernatant of LNCaP MM cells (Figs. 4G and EV5H), whereas their expression did not change in CAFs treated simultaneously with TLR3 inhibitor (Fig. EV5I,J), which seems to suggest, but not prove, a possible role of EV-transported TRPM8 RNA in shaping the functions of CAFs. Notably, supernatants from LNCaP MM cells promoted the expression of *CXCL1*, *CXCL10*, *IL1b*, and *IL6* genes in CAFs, whereas only the expression of *CXCL10* and *IL6* increased in NAFs (Fig. EV5K,L). A further substantial difference between LNCaP WT and MM tumors was the large extension of necrotic areas characterizing the MM xenografts (Fig. 4H,I). Nude mice have a partially impaired immune system due to the lack of the thymus gland and the almost complete absence of mature T cells. Since several other components of the immune system are retained in these mice, immunohistochemistry analyses for the antigen presenting cell (APC) marker MHC-II, macrophages marker Iba1, B-cell marker CD45R (B220), and natural killer (NK) marker NKp46 were performed to investigate the possible role *TRPM8*/TLR3/NF-kB circuit in anti-cancer immunity. APCs resulted highly abundant in both WT and MM tumors, but mainly distributed at the periphery (Figs. 4H and EV5M). Few B-cells and macrophages were observed in both WT and MM tumors. In contrast to APCs, B220+ and Iba1+ cells infiltrated tumor masses (Figs. 4H,J,K and EV5M). Finally, numerous NKp46+ cells preferentially distributed around the areas of necrosis were identified in both xenografts, with LNCaP MM tumors showing significantly higher numbers of infiltrating NKp46+ cells than LNCaP WT tumors (Figs. 4H,L and EV5M). Of note, the genes *NCR3LG1* and *BG6*, which encode for the two known ligands of NKp46 receptor used to bind target cells to be eliminated, showed higher expression in LNCaP MM than in LNCaP WT cells, while their expression was further reduced in LNCaP CAS cells (Fig. 4M).

## Discussion

The Toll-like receptor family consists of 13 different members, most of which recruit MyD88 and IRAK to activate downstream effectors upon recognition of PAMPs. Differently, TLR3 interacts with TRIF to signal the presence of short double-stranded RNAs

thus promoting a type 1 interferon (IFN) response along with NF-kB activation. In the context of prostate cancer, Toll-Like Receptor 3 (TLR3) has been shown to play an immune surveillance role in TRAMP mice (Chin et al, 2010). Activation of TLR3 by polyinosinic-polycytidylic acid (polyI:C) counteracts PCa growth in syngeneic transplanted mice through the recruitment of NK cells (Chin et al, 2010). In line with that, the secretion of extracellular vesicles containing the RNA encoding for the TRPM8 ion channel promotes a sterile inflammatory state purely dependent on TLR3 signaling in prostate cells. TRPM8-induced sterile inflammation significantly increases NK cell infiltration in PCa xenografts, which could account for the widespread necrosis favored by the raised expression in PCa cells of NKp46 ligands. Interestingly, the inflammatory wave propagates from the prostate cells to the fibroblasts, particularly CAFs, which on the one hand amplify the inflammatory signals while, on the other hand, produce and secrete a lower amount of type I collagen. Desmoplastic stroma is considered a clinical marker of poor prognosis in different types of solid tumors, including prostate carcinoma (Miles et al, 2019). CAFs can promote cancer cell survival by generating an immune suppressive environment and favor tumor progression by secreting high amounts of growth signals. They are also major contributors to the deposition of type I collagen, which plays distinct and controversial roles in tumorigenesis. Increased collagen density within the tumor extracellular matrix may modulate tumor cell metabolism in favor of glutamine instead of glucose to fuel the TCA cycle under nutrient starvation (Hsu et al, 2022; Morris et al, 2016). Increasing matrix stiffness promotes tumor progression through mechano-signals (Paszek et al, 2005; Provenzano et al, 2009; Mammoto et al, 2009; Calvo et al, 2013; Panciera et al, 2017), while concurrently suppressing tumor progression by mechanically limiting the spread of tumor cells (Bhattacharjee et al, 2021). Conversely, dense collagen type I matrix promotes invasiveness and metastasis once collagen fibers are linearized by physical stresses or enzymatic activities (Jia et al, 2019). Noteworthy, PSA has been recently linked to type I collagen degradation in PCa (Pellegrino et al, 2021). The amount of PSA detected within the prostate tissue (tPSA) inversely correlates with clinical and pathological parameters of PCa more precisely than the levels of circulating PSA. Functionally, tPSA hinders the propensity of cancer cells to invade the extracellular matrix by lowering the collagen type I/Integrin beta 1 signaling (Pellegrino et al, 2021). In addition to the direct influence on cancer cell survival and aggressiveness, recent works have shed light on the roles of collagen in cancer immunity. Dense type I collagen matrix has been shown to slow the migratory capacity of T cells, reduce tumor mass infiltration, and counteract

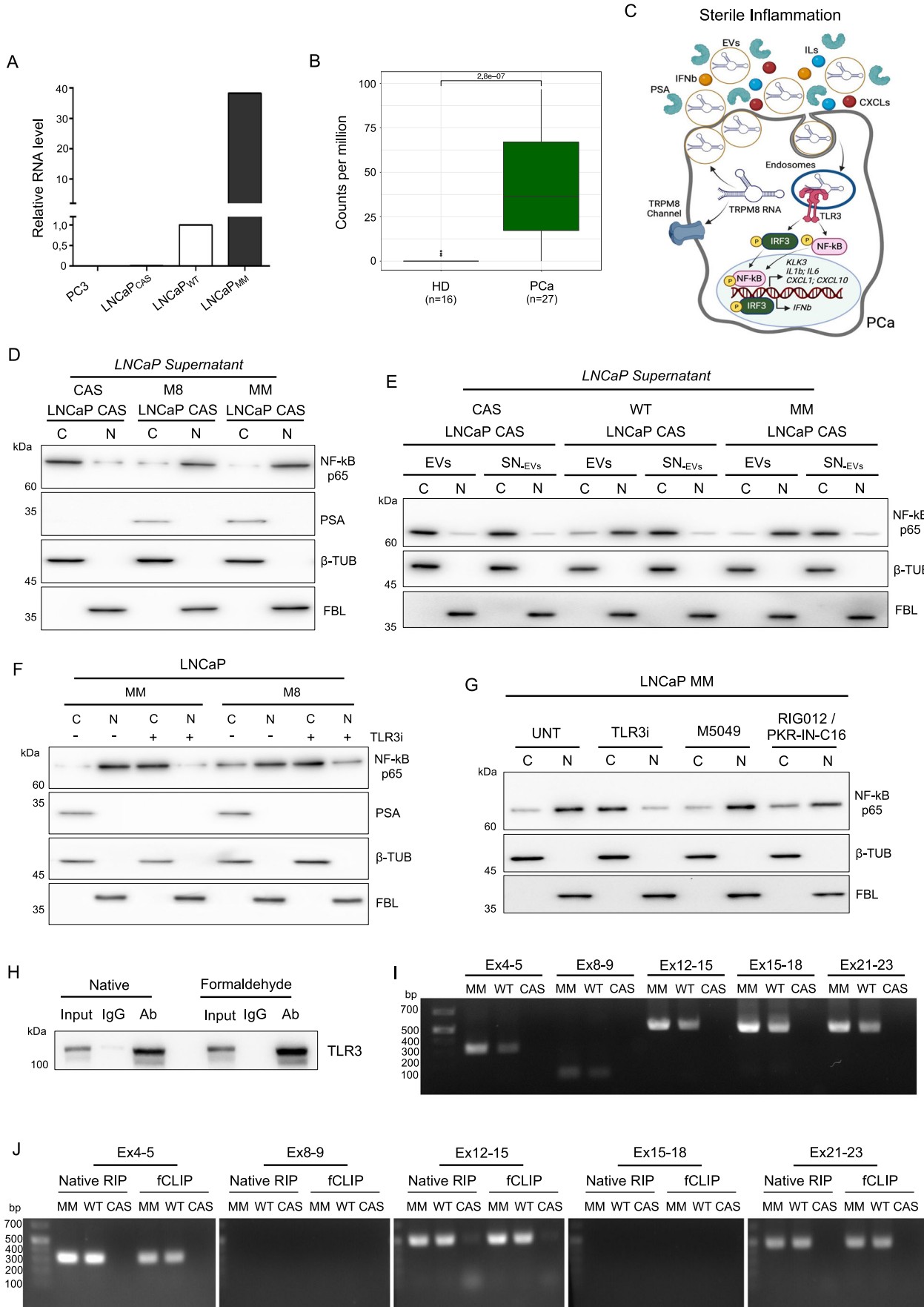

**Figure 3.  EVs secreted TRPM8 RNA triggers NF-kB activation via TLR3 signaling.**

(A) Relative amount of TRPM8 RNA carried by extracellular vesicles harvested from the supernatants of the indicated cell lines. TRPM8 not expressing (PC-3) and knocked-out (LNCaP CAS) prostate cancer cell lines were used as negative controls. (B) RNA-seq analysis of isolated extracellular vesicles (EVs) circulating in the blood of healthy donors (HD, $n = 16$) and PCa patients (PCa, $n = 27$) demonstrating the presence of TRPM8 RNA in EVs isolated from the blood of PCa patients. (C) Schematic representation of the TRPM8 RNA/TLR3 molecular circuit triggering sterile inflammation (*Created with BioRender.com*). (D) Nucleus/Cytosol fractionation and Western blot analysis of NF-kB p65 and PSA in LNCaP CAS cells conditioned with the supernatants of LNCaP (CAS, M8, and MM). β-Tubulin (β-TUB) and Fibrillarin (FBL) were used as markers of the cytosolic and nuclear fractions, respectively. (E) Nucleus/Cytosol fractionation and Western blot analysis of NF-kB p65 in LNCaP CAS cells conditioned with EVs and the EVs-depleted supernatants ($SN_{-EVs}$) of LNCaP (CAS, WT, and MM). β-Tubulin (β-TUB) and Fibrillarin (FBL) were used as markers of the cytosolic and nuclear fractions, respectively. (F) Nucleus/Cytosol fractionation and Western blot analysis of NF-kB p65 and PSA in LNCaP MM and M8 cell lines in the presence or absence of TLR3/dsRNA Complex Inhibitor (TLR3 inhibitor; 20 μM, 24 h). β-Tubulin (β-TUB) and Fibrillarin (FBL) were used as markers of the cytosolic and nuclear fractions, respectively. (G) Nucleus/Cytosol fractionation and Western blot analysis of NF-kB p65 in LNCaP MM cells in the presence of TLR3 Inhibitor (20 μM, 24 h), TLR7/8 inhibitor M5049 (1 μM, 6 h), or the combination of RIG1 inhibitor RIG012 (2 μM, 6 h) and PKR inhibitor C16 (2 μM, 6 h). Untreated LNCaP MM cells served as positive control for NF-kB nuclear localization. β-Tubulin (β-TUB) and Fibrillarin (FBL) were used as markers of the cytosolic and nuclear fractions, respectively. (H) Western blot analysis of TLR3 immunoprecipitation from LNCaP under native and formalin fixed (0.1% f.c., 10 min) conditions. Input line shows 10% of total protein extract. (I,J) PCR analysis of total (I) and TLR3-bound (J) TRPM8 transcript in LNCaP MM, WT and CAS cell lines with sets of primers spanning different exons (Ex) of the coding sequence of TRPM8 RNA. Data Information: In (A) data are presented as results of a single experiment (replicates are presented in Fig. EV3A). In (B) box-plots elements indicate the median (center line), upper and lower quartiles (box limits). Whiskers extend to the most extreme value included in 1.5 × interquartile range. (Two-sided Wilcoxon rank-sum test). Source data are available online for this figure.

the killing activity of CD8+ cytotoxic-T and NK cells (Hartmann et al, 2014; Rygiel et al, 2011; Hörner et al, 2019; Nicolas-Boluda et al, 2021; Tabdanov et al, 2021; Kuczek et al, 2019; Sun et al, 2021). Therefore, we cannot exclude the possibility that decreased type I collagen deposition by fibroblasts and increased degradation promoted by higher levels of tPSA in PCa xenografts undergoing TRPM8/TLR3-dependent sterile inflammation may contribute to the greater tumor infiltration and killing activity of NK cells in these tumors. However, changes in fibroblast populations and extracellular matrix structure associated with the tumor process (Mayorca-Guiliani et al, 2017; Affo et al, 2021) may influence cancer immunity by following organ-specific rules, as suggested by the immune suppressive effect of genetic deletion of type I collagen in α-Sma + myofibroblasts described by Chen and colleagues in mouse models of pancreatic cancer (Chen et al, 2021).

Overall, this work defines a novel type of sterile inflammation with critical roles in prostate disease and tumorigenesis (Fig. 5). Identification of molecular mechanisms responsible for aseptic chronic prostatitis may lead to tailored anti-inflammatory treatments (e.g., Takinib), while a better understanding of the regulatory pathways controlling PSA can contribute to a more accurate interpretation of tumor response to therapy. By promoting androgen-independent transcription of the *KLK3* gene via NF-kB and intratumor sequestration of PSA shaping the tumor micro-environment, sterile inflammation induced by PCa cells secretion of TRPM8 RNA in EVs may introduce a further layer of complexity in the evaluation of PSA levels or PSA derivatives affecting our ability to monitor the disease (PSA doubling time, PSA velocity, biochemical recurrence, biochemical response/progression). Finally, by advancing our knowledge of the molecular circuits governing innate and adaptive immunity, this work provides important insights towards the definition of new therapeutic routes in oncology focused on TLR3 targeting (Le Naour et al, 2020).

# Methods

## Cell culture

LNCaP (#CRL-1740), RWPE-1 (#CRL-11609) and PC-3 (#CRL-1435) cell lines were commercially obtained from the ATCC in July

2015. Plasmid construction and lentiviral transduction for TRPM8 overexpression and TRPM8 Knock Out, used to achieve the lines RWPE-1 M8, LNCaP M8, RWPE-1 CAS, and LNCaP CAS, were established as previously described (Alaimo et al, 2020). To obtain the cell lines LNCaP MM and RWPE-1 MM, containing the TRPM8 transcript (MM) that is unable to translate the channel, the TRPM8 cDNA was mutated to convert methionine 1 and 10 in leucine, PCR amplified and cloned into the NotI- and BsrGI-digested pAIB lentiviral vector. The final product was sequence verified. Stable cell lines were generated by lentiviral transduction in LNCaP CAS and RWPE-1 CAS followed by selection with blasticidin (10 μg/ml, InvivoGen) as previously described (Alaimo et al, 2020). Human prostate fibroblast cell lines were derived from radical prostatectomy. Prostate cancer patients were enrolled with written informed consent on a protocol approved by the Ethical Committee of the Molinette Hospital, Turin (Rep. Int. 0009136). Tissue samples from both normal and cancerous parts as per pathologist analysis were collected in Tissue Storage Buffer (Miltenyi Biotec) under sterile conditions. After extensive washing in PBS and medium containing 2X Pen/Strep, samples were chopped into small pieces, washed twice in complete medium (DMEM supplemented with 10% FBS, glutamax, 100 U/mL Pen/Strept; cDMEM) and digested O/N with 1 mg/ml of collagenase II (Sigma-Aldrich) at 37 °C, with shaking, followed by cDMEM washing, digestion with 0.05% Trypsin/EDTA and DNase I (Roche), 25 mg/ml. The pellet was suspended in cDMEM 20% FBS and seeded at 37 °C, 5% $CO_2$, and 95% humidity. The derived cell populations were analyzed by FACS highlighting $CD90^+CD45^-CD31^-EpCAM^-$ cells EpCAM (Milteny Biotec, #130113-826), CD45 (Milteny Biotec, #130-110769), CD31-VioBlue (Milteny Biotec, #130-119-980), CD90-APC (Milteny Biotec, #130-114-903). Cultures were passaged 1:3 after reaching confluence and used up to passage 4. For immortalization, passage 2 fibroblasts were transduced with the pBABE-puro-hTERT vector (Addgene, #1771), followed by two rounds of 48 h selection with puromycin (2 μg/ml) to obtain stable cell lines and subsequent regular passages. Cells were cultured in a humidified incubator at 37 °C and 5% $CO_2$ and maintained according to the manufacturer's instructions. In experiments where the effects of the androgens were assessed, LNCaP cells were grown in RPMI 1640 without phenol red supplemented with 10% charcoal-stripped FBS

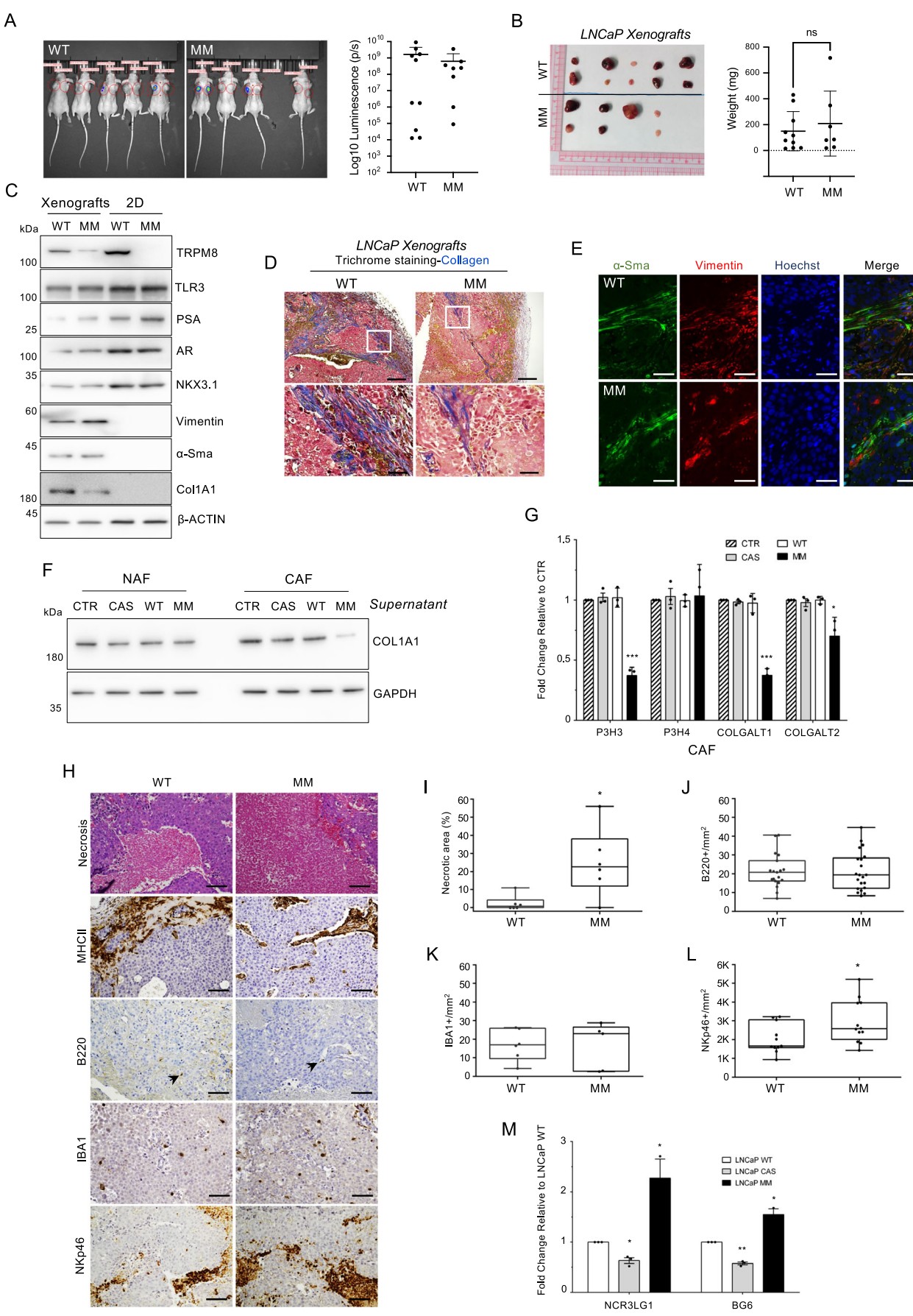

**Figure 4. TRPM8/TLR3 inflammation promotes Natural Killer cells infiltration and tumor necrosis.**

(**A**) Images showing in vivo bioluminescence quantification of subcutaneous xenografts obtained with LNCaP WT luc ($n = 10$) and LNCaP MM luc ($n = 7$) 8 weeks after cells injection. (**B**) Images (left) and weight (right) of LNCaP WT luc- and LNCaP MM luc-derived xenografts 8 weeks after cells injection. (**C**) Western blot analysis of proteins extracted from FFPE sections of LNCaP WT and LNCaP MM xenografts and in vitro cultured LNCaP WT and LNCaP MM cell lines. $\beta$-Actin was used as loading control. (**D**) Trichrome staining showing lower deposition of collagen (blue staining) in xenografts generated by LNCaP MM cells ($n = 3$) compared to those obtained with LNCaP WT ($n = 3$). Scale bars, 100 µm. (**E**) Immunofluorescence analysis of alpha-Smooth muscle actin ($\alpha$-Sma) and Vimentin in FFPE sections of LNCaP WT and LNCaP MM xenograft. Scale bars, 100 µm. (**F**) Immunoblotting analysis of COL1A1 in NAF and CAF cells derived from human prostate tissues and conditioned with the supernatants of both LNCaP (CAS, WT, MM) and RWPE-1 (CAS, WT, MM) cell lines, respectively. GAPDH was used as loading control. (**G**) RT-qPCR analysis of the indicated genes of the metabolism of type I collagen in CAF cells. (**H**) Cytochemistry and immunohistochemistry analyses of LNCaP WT and MM xenografts. Representative images of hematoxylin and eosin (H&E) staining (necrosis) and IHC analyses for MHCII, CD45R (B220), IBA1 and NKp46 in tissue sections from PDXs. Scale bars, 200 µm. (**I,L**) Quantitation of necrotic areas and immune cells infiltrate in multiple sections of LNCaP WT ($n = 3$) and LNCaP MM ($n = 3$) xenografts. (**M**) RT-qPCR analysis of NCR3LG1 and BG6, two known ligands of NKp46 receptor, in LNCaP WT, CAS and MM cell lines. Data Information: In (**A,B**) data are presented as mean ± SD. Not statistically significant (ns). (Student's $t$-test). In (**G,M**) data are presented as mean ± SD of $n = 3$ independent biological replicates. *$P \leq 0.05$; **$P \leq 0.01$; ***$P \leq 0.001$. (Two-tailed Student's $t$-test). In (**I–L**) box-plots elements indicate the median (center line), upper and lower quartiles (box limits). Whiskers extend from the minimum to the maximum. *$P \leq 0.05$ (Student's $t$-test). Source data are available online for this figure.

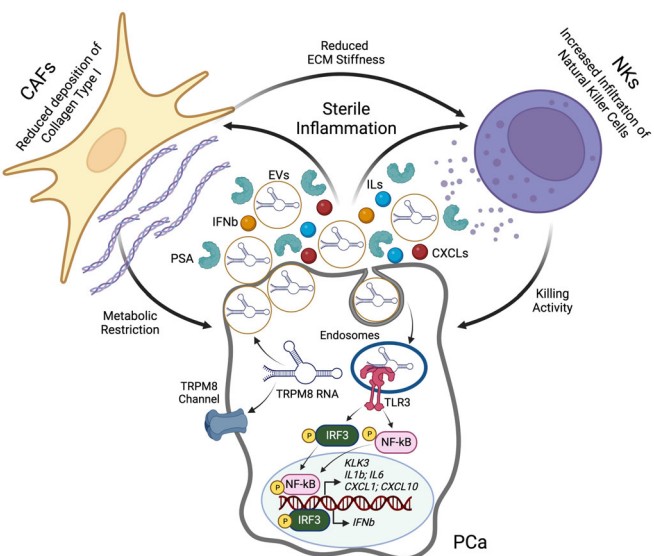

**Figure 5. Sterile inflammation and cancer immunity.**

Molecular and cellular consequences of sterile inflammation triggered by TRPM8 RNA in the prostate cancer microenvironment (*Created with BioRender.com*).

(Hyclone, Celbio) for 72 h and then treated as indicated. All cell lines were routinely tested for Mycoplasma contamination (MycoAlert, Lonza).

## Growth assay

LNCaP cells were plated in 12-well plate ($1 \times 10^4$ cells/well) in triplicates. Afterward, cells were washed with PBS, fixed with 10% formalin (Sigma), and stained with 0.1% Crystal Violet (Sigma) in 20% methanol solution for 30 min. Cells were washed with dH20, dried, and Crystal Violet was extracted with 10% acetic acid for 30 min. Finally, absorbance was measured at 595 nm.

## Migration assays

Transwell cell migration assays were performed using the CHEMICON® QCM™ fluorimetric Cell Migration Assay Kit (Merck-Millipore, ECM 509), following the manufacturer's

instructions. Cells were serum starved for 24 h, then 300 µl of $5 \times 10^4$ cells supplemented with 0,1% BSA were seeded into the migration chamber. A total of 500 µl of complete RPMI medium (with 10% FBS as a chemoattractant for LNCaP cells) was added to the feeder tray. The migration chamber plate was incubated in the cell detachment solution for 30 min at 37 °C, then 75 µl of Lysis Buffer/Dye Solution was added to the well and incubated for 15 min at RT, and fluorescence was measured with an Infinite M-200 Pro fluorescence plate reader (Tecan) using 480/520 nm a filter set. Transwell migration experiments were run in triplicates, quantified and reported graphically as bar charts. For the scratch motility experiments, cells were cultured to confluence in six-well plates at 37 °C. A 200 µl pipette tip was used to scratch the cells monolayer across the wells and images were captured using a Leica DFC 450 C microscope after the scratch (0 h) and after 24, 48, and 72 h. The migration rates of the LNCaP cells were estimated using ImageJ software (ImageJ 1.46r NIH). Experiments were performed in triplicates.

## Flow cytometry analysis

For cycle analysis, LNCaP WT and LNCaP MM were harvested, collected and washed in ice-cold PBS. Cells were fixed in ice-cold EtOH 70% at 4 °C for 1 h, then washed in PBS. Afterward, cell pellet was incubated in 100 µl PBS containing 0.5 µg/ml RNaseA (Life Tech) for 30 min at 37 °C. Cells were incubated with 100 µL propidium iodide (50 µg/mL, Life Tech) in PBS for 30 min at room temperature before the analysis. Cell death was determined with Annexin-V-FITC and propidium iodide staining according to manufacturer's instructions (Annexin V FITC Kit, Miltenyi Biotec). For FACS analysis a BD FACSymphony™ A1 Cell Analyzer (BD Biosciences) was used, and data were analyzed and quantified with FlowJo software (Treestar). FACS analysis were performed in three independent biological replicates; representative data are shown.

## Quantitative RT-qPCR and end-point PCR

RT-qPCR and end-point PCR amplification were carried out as described in ref. (Lorenzoni et al, 2022). Analyses were performed with at least three independent biological replicates unless stated in the figure legend, and results were normalized to GAPDH mRNA levels. The specific primer sequences used are reported in Table EV1.

## Digital PCR

cDNA was generated from total cellular RNA (300 ng) according to iScript Select cDNA Synthesis Kit (Bio-Rad; #1708896) for oligo(dT) priming. ddPCR Supermix for Probes (Bio-Rad, #1863024) was used in sample preparation for droplets generation in the QX200 Droplet Digital PCR System (Bio-Rad Laboratories, Inc.) and PCR reactions were run on a C1000 touch thermal cycler (Bio-Rad, Hercules, CA, USA). Probes for TRPM8 (dHsaCPE5026734) and ACTIN BETA (dHsaCPE5190200) were from Bio-Rad Laboratories Inc. PCR plates were analyzed on a Bio-Rad QX200 droplet reader (Bio-Rad Laboratories, Inc.) and analysis of ddPCR data was performed by using QX Manager Software, standard edition (Bio-Rad Laboratories, Inc.).

## Protein analysis

Nuclear/cytoplasmic fractionation was performed using NE-PER Nuclear and Cytoplasmic Extraction Kit (Life Tech) according to the manufacturer's instructions. To extract proteins from Formalin-fixed paraffin-embedded (FFPE) we used the FFPE Protein Extraction Solution Kit (Agilent) following the manufacturer's instructions. Extracted protein samples were separated on 10–12% SDS-PAGE gels and transferred to a Polyvinylidene fluoride (PVDF, Amersham™ Hybond™, Fisher Scientific) membrane, as previously described (Genovesi et al, 2022). Western blots were performed in at least three independent biological replicates; representative data are shown. The primary antibodies used are provided in Table EV2. The amount of PSA in the supernatants was determined through the Elecsys total PSA (Roche Diagnostic) using the Cobas E801 immunoassay analyzer (Roche Diagnostic).

## Immunoprecipitation

Cells were washed with ice-cold PBS, incubated with 0.1% formaldehyde crosslinking solution for 10 min, quenched with 150 mM glycine and washed again with PBS. Each experiment was also carried out under native conditions. Cells were lysed in RIPA buffer plus protease inhibitor cocktail and centrifugated at $18,000 \times g$ at 4 °C for 20 min. Supernatants were collected and incubated with 1 µg of anti-TLR3 antibody (Novus, NBP2-24875) or 1 µg of Control IgG (BioLegend, 400940) O/N at 4 °C. Subsequently, Protein A sepharose beads (10 µL, Invivogen) were added to the complex and incubated at 4 °C for 3 h. Beads were washed twice with ice-cold PBS and bound proteins were eluted by heating with 5X Laemmli buffer at 95 °C for 10 min. Proteins immunoprecipitated were probed by WB for TLR3 with an anti-TLR3 antibody (Novus, NB100-5657).

## RIP and fCLIP

Formaldehyde crosslinking immunoprecipitation (fCLIP) and Native RIP experiments were performed according to the protocol described in (Kim and Kim, 2019) with minimal modifications. Briefly, formaldehyde-treated and untreated cells were washed, harvested, and lysed in CLIP Wash buffer containing protease inhibitor cocktail and RNase inhibitor (SUPERase In RNase Inhibitor, Invitrogen). Cell lysates were centrifuged, and supernatants incubated with 10 µl of anti-TLR3 antibody (1 µg, Novus,

NBP2-24875) – Protein A sepharose bead slurry for 4 h at 4 °C. Following the incubation, beads were washed and the elution of the TLR3 – RNA complexes and the recovery of RNA were performed as described (Kim and Kim, 2019). RNA was extracted (Acid-Phenol:Chloroform, Invitrogen), treated with DNAase I, and reverse-transcribed. qPCR analysis was performed with sets of primers for TRPM8, GAPDH and Y3 (see Table EV1). The obtained amplicons were loaded into an agarose gel, run and then purified for sequencing (see Appendix Fig. S2).

## Chemicals and drugs

Takinib, TLR3/dsRNA Complex Inhibitor, and WS-12 were purchased from Sigma-Aldrich, AMTB from Alomone Labs, and Enzalutamide from Cayman chemicals, Staurosporine from Sigma, M5049 (Enpatoran) and Poly(I:C) from Invivogen, RIG012 from Axon Medchem, PKR-IN-C16 from Selleckchem. All drugs were maintained as stock solutions and stored at −80 °C or −20 °C. In each experiment, the same volume of solvent used for tested drugs and chemicals was added to the control solution.

## Small interfering RNA against TRPM8

For gene knockdown of TRPM8, LNCaP cells were transfected with non-targeting (siCTR) or TRPM8 targeting (siRNA1 or siRNA2) small interfering RNA molecules (obtained from Ambion, Life Tech), following the approach described in (Alaimo et al, 2020). Downregulation of the expression of TRPM8 in LNCaP cells was confirmed using RT-qPCR and Western blot analyses 48 h after transfection.

## Extracellular vesicles isolation and RNA profiling

Prostate cancer patients' blood samples were collected as part of the funded PRIME project (PI and program coordinator, prof. F. Demichelis – University of Trento), aiming to build minimally invasive multimodal assays for advanced prostate cancer clinical management. Patients were prospectively enrolled with written informed consent on a protocol approved by the Ethics Committee of the Santa Chiara Hospital in Trento (Rep. Int. 2562 of February 12, 2019; coordinating clinical center) and subsequently by the Ethics Committee of the Istituto Oncologico Veneto (IOV), Padova. Patients included in the study ($n = 27$) had CRPC diagnosis at IOV, started Enzalutamide treatment within 120 days from the diagnosis, and had no prior chemotherapy. Whole blood samples were collected from patients in K2EDTA tubes (Vacuette) at baseline. Plasma samples from healthy donors ($n = 16$) were purchased from Precision Medicine Group, LLC. Single plasma aliquots were processed using the ONCE protocol (https://doi.org/10.1101/2023.03.02.530645) to concomitantly extract extracellular vesicles (EV) and circulating nucleic acids. EV were isolated by ultracentrifugation on an Optima MAX-XP ultracentrifuge (Beckman Coulter) equipped with a TLA55 rotor as previously reported (Thery C, Amigorena S, Raposo G, Clayton A. Isolation and characterization of exosomes from cell culture supernatants and biological fluids. Curr Protoc Cell Biol 2006). RNA was purified with qEV RNA Extraction Kit (IZON) according to the manufacturer's instructions and treated with RNAse-free DNAse I Kit (Norgen Biotek). RNA libraries were prepared according to the

manufacturer's instructions using Clontech Takara SMARTer smRNA-Seq Kit. Sequencing was performed on Illumina Novaseq 6000 by the NGS facility at the University of Trento. Fastq files were trimmed with Cutadapt (Martin, 2011) according to the library preparation kit manual to remove artificial poly-A and SMART template-switching bases. Reads were aligned to hg38 using STAR v2.7.2b (Dobin et al, 2013). RNA fold analysis was performed using the RNAfold function of ViennaRNA 2.5.1 (Lorenz et al, 2011). Pileup files were generated using Pacbam (Valentini et al, 2019). Statistical testing and data visualization were performed using R (https://www.R-project.org) and ggplot package (Wickham, 2016).

EVs isolation from the supernatants of PC-3, LNCaP CAS, LNCaP WT and LNCaP MM cell cultures was carried out by ultracentrifugation as described in (Di Vizio et al, 2012). Total RNA was extracted from using QIAzol reagent (QIAGEN) according to the manufacturer's instructions and retrotranscribed into cDNA. Gene expression analysis was performed via qPCR with specific sets of primers (Table EV1).

## TRPM8 RNA structure prediction

The secondary structures of human and mouse TRPM8 mRNA were predicted with RNAfold from the ViennaRNA package v2.5.0 (Lorenz et al, 2011) using the -p and --MEA options. The resulting structures were then plotted with RNAplot from the same package, using the RNApuzzler algorithm (-t 4 option) and custom postscript code to highlight potential TLR3 binding sites (--pre option). Base-pair probabilities were eventually added to the secondary structure plot using the relplot.pl script (also from the ViennaRNA package). The mountain plot of base-pair probabilities was obtained with the mountain.pl script of the ViennaRNA package v2.5.0.

## LNCaP WT and MM cells transduction with mutant click beetle luciferase

For the preparation of the lentivirus expressing the mutant click beetle luciferase, HEK-293T cells were transfected with 4 µg of envelope expressing plasmid pMD2.G (Addgene #12259), 7.5 µg of lentiviral vector packaging plasmid psPAX2 (Addgene #12260) and 12.5 µg of ATG-2460 containing the mutant click beetle luciferase. ATG-2460 was a gift from Keith Wood (Addgene #108713). LNCaP WT and MM cells were transduced with EF1-CBR2opt-T2A-copGFP (ATG-2460) lentivirus, and FACS sorted for GFP expression at confluency.

## Animal studies

Animal experiments were conducted according to Bern cantonal guidelines under the license BE 67/20. Mice had unrestricted access to food and fresh water and were housed in max 5 animals per cage. Male BALB/c nude mice were used for subcutaneous transplantation experiments. Mice were anesthetized with a cocktail of medetomidin (Dorbene) 1 mg/kg, midazolam (Dormicum) 10 mg/kg, and fentanyl 0.1 mg/kg. Anesthesia was reversed by subcutaneous injection with atipamezol (Revertor®) 2.5 mg/kg and flumazenil (Anexate®) 0.5 mg/kg. Two experimental groups of 5 mice stratified by weight were generated and randomly assigned to

the injection of LNCaP WT or LNCaP MM. Two million LNCaP WT or LNCaP MM cells were injected in 50 µl of Geltrex subcutaneously in the scapular area. Mouse weight and tumor size were monitored once a week, tumor size was tracked both by bioluminescence measurement and caliper measurement (once tumor reached palpable size). Whole body bioluminescent imaging (BLI) was performed using IVIS Spectrum CT (Xenogen/Perkin Elmer). Mice were anesthetized using isoflurane and injected intraperitoneally with 100 µl of 2 mg D-luciferin (Per bio Science Nederland B.V.). Analyses for each injected site were performed after definition of the region of interest and quantified with Living Image 4.2 (Caliper Life Sciences). Electrochemiluminescence Immunoassay "ECLIA" on cobas® e 801 was performed to measure total (free and complexed) prostate-specific antigen (tPSA) in the mouse plasma. Results are indicated in µg/L.

## Immunofluorescence analysis

Cells were seeded on coverslips, fixed with 4% paraformaldehyde (PFA) and permeabilized in 5% FBS with 0.1% Triton X-100. Cells were incubated with NF-kB primary antibody (see Table EV2) overnight at 4 °C. After washing, cells were incubated with Alexa Fluor conjugated secondary antibody and counterstained with DAPI.

## Histology and immunohistochemistry analysis

Tissue specimens and RWPE-1 cell pellets were fixed in 4% PFA, dehydrated, and embedded in paraffin wax. Formalin-fixed paraffin-embedded (FFPE) sections were cut at a thickness of 5 µm according to standard procedures and stained with hematoxylin and eosin (HE, Merck) for histologic analysis. For direct visualization of collagen fibers and histochemical evaluation of collagen deposition, sections were analyzed with the Masson's Trichrome staining (Masson Trichrome kit, Bio-Optica). Immunofluorescence staining with anti-α-SMA and anti-Vimentin antibodies was performed according to the previous study (Cambuli et al, 2022). Immunohistochemical staining with anti-PSA, anti-MHCII, anti-NKp46/NCR1 and anti-IBA1 antibodies was achieved as previously described (Genovesi et al, 2022; Cambuli et al, 2022). Briefly, antigens were retrieved with citrate buffer (pH 6.0; Vector Lab), then slides were incubated in blocking solution (5% FBS + 0.1% Triton-X in PBS), before proceeding to staining with primary antibodies, at 4 °C, overnight. After washes, sections were incubated with biotin-conjugated secondary antibodies and the immunoreaction visualization was performed according to the procedure mentioned previously (Alaimo et al, 2020; Lorenzoni et al, 2022). The IHC analysis with B220 rat monoclonal antibody clone RA3-6 was performed at the Division of Pathology (San Raffaele Hospital, Milan, Italy). Sections were counterstained with Hematoxylin. All the slides were reviewed independently by three trained pathologists (MB, GB and FGC). Images were acquired using an Axio Imager M2 (Zeiss). Primary antibodies are listed in Table EV2. Images of whole xenograft sections were acquired as MosaiX with a Zeiss Axio Imager M2 up-right wide-field microscope equipped with a Zeiss AxioCam MRc color camera, using an EC Plan-Neofluar 10x/0.3 numerical aperture (NA) objective and the Zeiss AxioVision Software. Quantification of collagen was performed with QuPath version v.0.3.2 (Bankhead

et al, 2017). The algorithm was based on a pixel classifier and was trained on multiple regions of different tissue sections with dedicated annotations. Detection of PSA+ and NKp46+ cells was conducted using QuPath's built-in "Positive cell detection" (QuPath version v.0.3.2) (Bankhead et al, 2017).

## Gene expression analysis

The gene expression data from RWPE-1 and LNCaP cell lines were downloaded from the Gene Expression Omnibus (GEO) (Barrett et al, 2012) together with the corresponding metadata using the GEOquery R package (Davis and Meltzer, 2007). Probes were annotated to their corresponding Gene Symbols, and in the case of multiple probes mapping to the same gene, only the most highly expressed probe (on average) was retained. Gene lengths for RPKM normalizations of GSE141323, GSE73784, GSE64529, and GSE117922 datasets (provided as counts) were obtained from the EDASeq package (Risso et al, 2011). Data were log2 transformed with an offset of 1. The heatmap was plotted with the pheatmap package, scaling gene expression by row. All the analyses were done with R 4.0.3.

## Statistical analysis

Data are presented as mean $\pm$ SD of at least three biological replicates and Two-tailed Student's *t*-test was used to assess statistical significance, unless differently specified. $P < 0.05$ was set as threshold for significant results. All statistical data analysis was performed using GraphPad Prism 8.0.

# Data availability

All data needed to evaluate the conclusions in the paper are presented in the paper and/or Supplementary Materials. The GEO ids of the selected datasets are: Data ref: GSE98898 (Luo et al, 2017), Data ref: GSE112007 (Li et al, 2019), Data ref: GSE128399 (Kamdar et al, 2019), Data ref: GSE141323 (Koirala et al, 2020), Data ref: GSE73784 (Taberlay et al, 2016), Data ref: GSE64529 (Metzger et al, 2016), Data ref: GSE162947 (Xu et al, 2021), Data ref: GSE117922 (Mazzu et al, 2020), Data ref: GSE89226 (Mu et al, 2017), Data ref: GSE55030 (Zhao et al, 2016), Data ref: GSE80450 (Paltoglou et al, 2017), Data ref: GSE83547 (Flaig et al, 2017), Data ref: GSE85556 (Shukla et al, 2017). No additional large-scale data amenable to public repository deposition were generated in this study.

# Peer review information

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

## Acknowledgements

We thank current and former members of the Lunardi laboratory for experimental support and advice. We are grateful to all the staff at the CIBIO core facilities for their help. Department CIBIO Core Facilities are supported by the European Regional Development Fund (ERDF) 2014–2020. Furthermore, we thank all the staff at the Department of Histopathology (S. Chiara Hospital, Trento, Italy) for their technical support with the histological work. Illustrations were created with Inkscape and BioRender.com. This work was supported by the Giovanni Armenise-Harvard Foundation (Career Development Award), the Italian Ministry of University and Research (PRIN 20174PLLYN), the Associazione Italiana per la Ricerca sul Cancro (AIRC-IG 27893), the Lega Italiana Lotta ai Tumori (LILT-Bolzano), and by the core funding from the Department CIBIO to AL; by the Italian Ministry of University and Research (PRIN 20174PLLYN) and Associazione Italiana per la Ricerca sul Cancro (AIRC-IG 24851) to VP; by the Italian Ministry of University and Research (PRIN 20174PLLYN) to IMB; by the Italian Ministry of University and Research (PRIN 20174PLLYN) to MGP; by the Associazione Italiana per la Ricerca sul Cancro (AIRC MFAG 2017-ID 20621) to AR; by the Associazione Italiana per la Ricerca sul Cancro (AIRC-IG 25978) to RT; by AIRC Foundation (ID 22792) and Cancer Research UK (CRUK, ID A26822) Accelerator Award 2018 to FD; by Krebsliga Schweiz (KFS-4960-02-2020) and Swiss National Science Foundation Sinergia grant (SNSF 31003A_169352) to MKDJ; by Fondazione Pezcoller to OC; by the University of Trento (Starting Grants Young Researchers 2019) to AA Individual fellowships were awarded from the Fondazione Umberto Veronesi (FUV 2016) to AA, and from the University of Trento (Ph.D. fellowship) to DDF.

## Author contributions

**Alessandro Alaimo**: Conceptualization; Data curation; Formal analysis; Validation; Investigation; Visualization; Methodology; Writing—original draft; Project administration; Writing—review and editing. **Sacha Genovesi**: Data curation; Formal analysis; Validation; Investigation; Methodology. **Nicole Annesi**: Data curation; Formal analysis; Validation; Investigation. **Dario De Felice**: Data curation; Validation; Investigation; Methodology. **Saurav Subedi**: Data curation; Formal analysis; Validation; Investigation; Methodology. **Alice Macchia**: Data curation; Validation; Investigation; Methodology. **Federico La Manna**: Data curation; Formal analysis; Investigation; Methodology. **Yari Ciani**: Data curation; Software; Validation; Investigation; Methodology. **Federico Vannuccini**: Data curation; Validation; Investigation; Methodology. **Vera Mugoni**: Data curation; Formal analysis; Validation; Investigation; Methodology. **Michela Notarangelo**: Formal analysis; Investigation; Methodology. **Michela Libergoli**: Formal analysis; Investigation; Methodology. **Francesca Broso**: Formal analysis; Investigation; Methodology. **Riccardo Taulli**: Investigation; Methodology. **Ugo Ala**: Investigation; Methodology. **Aurora Savino**: Software; Formal analysis; Validation; Investigation; Methodology. **Martina Cortese**: Formal analysis; Investigation; Methodology. **Somayeh Mirzaaghaei**: Resources; Investigation; Methodology. **Valeria Poli**: Resources; Visualization; Methodology. **Ian-Marc Bonapace**: Resources; Visualization; Methodology. **Mauro Papotti**: Resources; Visualization; Methodology. **Luca Molinaro**: Resources; Visualization; Methodology. **Claudio Doglioni**: Resources; Visualization; Methodology. **Orazio Caffo**: Resources; Visualization; Methodology. **Adriano Anesi**: Resources; Formal analysis; Methodology. **Michael Nagler**: Resources; Formal analysis; Methodology. **Giovanni Bertalot**: Resources; Formal analysis; Visualization; Methodology. **Francesco Giuseppe Carbone**: Resources; Formal analysis; Investigation; Visualization; Methodology. **Mattia Barbareschi**: Resources; Visualization; Methodology. **Umberto Basso**: Resources; Formal analysis; Methodology. **Erik Dassi**: Resources; Data curation; Formal analysis; Validation; Methodology. **Massimo Pizzato**: Resources; Validation; Visualization; Methodology. **Alessandro Romanel**: Resources; Software; Formal analysis; Validation; Visualization; Methodology. **Francesca Demichelis**: Resources; Data curation; Software; Formal analysis; Validation; Visualization; Methodology. **Marianna Kruithof-de Julio**: Resources; Formal analysis; Validation; Investigation; Visualization; Methodology. **Andrea Lunardi**: Conceptualization; Data curation; Formal analysis; Supervision; Funding acquisition; Validation; Investigation; Visualization; Methodology; Writing—original draft; Writing—review and editing.

## Disclosure and competing interests statement

The authors declare no competing interests.

# Expanded View Figures

**Figure EV1.  TRPM8 favors KLK3 expression in prostate cells.**

(**A**) Immunoblot analysis showing expression of TRPM8, PSA and the other indicated proteins in LNCaP wild type (WT) and stably overexpressing TRPM8 (M8). GAPDH was used as loading control. (**B**) List of genes ranked by correlation coefficient whose expression in cancer correlates with *TRPM8* expression (TCGA, cBioportal). (**C,D**) RT-qPCR (**C**) and Western blot (**D**) analyses of *TRPM8* and *KLK3* (PSA) expression in RWPE-1 wild type (WT) and in three independent clones of RWPE-1 stably overexpressing TRPM8 (M8.1; M8.2; M8.3). GAPDH was analyzed after stripping of the primary and secondary antibodies used for PSA. (**E**) RT-qPCR analysis of RWPE-1 and RWPE-1 M8 cells treated with 100 nM Enzalutamide for 48 h. The Ct values associated with *KLK3* expression in RWPE-1 (34.33, 34.56, 34.72) confirmed the negligible expression of *KLK3* gene in these cells. Data Information: In (**C,E**) data are presented as mean ± SD of $n = 3$ independent biological replicates. *$P \le 0.05$; **$P \le 0.01$; ***$P \le 0.001$; ns, not statistically significant (Two-tailed Student's *t*-test). Source data are available online for this figure.

▶

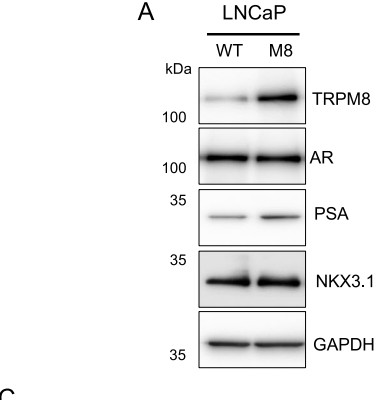

A

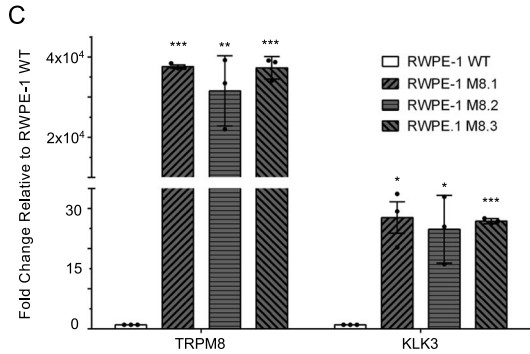

C

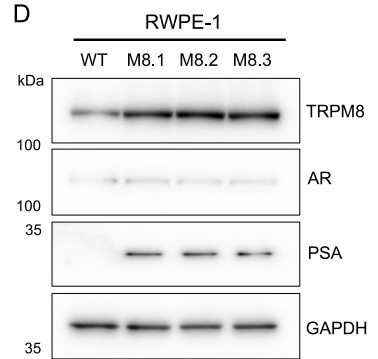

D

B

*TCGA-PanCan*

| Correlated Gene | Cytoband | Spearman's Correlation | p-Value | q-Value |
|---|---|---|---|---|
| SLC4A4 | 4q13.3 | 0.560 | 1.40e-41 | 2.80e-37 |
| CCNG2 | 4q21.1 | 0.555 | 8.20e-41 | 8.20e-37 |
| ENDOD1 | 11q21 | 0.520 | 3.73e-35 | 2.49e-31 |
| MPC2 | 1q24.2 | 0.493 | 3.28e-31 | 1.64e-27 |
| ELOVL7 | 5q12.1 | 0.491 | 4.94e-31 | 1.97e-27 |
| MBOAT2 | 2p25.1 | 0.490 | 7.50e-31 | 2.50e-27 |
| ABCC4 | 13q32.1 | 0.473 | 1.25e-28 | 3.59e-25 |
| VPS54 | 2p15-p14 | 0.470 | 3.28e-28 | 8.20e-25 |
| RPS6KA3 | Xp22.12 | 0.464 | 1.93e-27 | 4.29e-24 |
| SLC9A2 | 2q12.1 | 0.462 | 4.12e-27 | 8.23e-24 |
| AADAT | 4q33 | 0.460 | 5.59e-27 | 9.45e-24 |
| GLYATL1 | 11q12.1 | 0.460 | 5.67e-27 | 9.45e-24 |
| GABRG3 | 15q12 | 0.455 | 2.28e-26 | 3.50e-23 |
| MSL3P1 | 2q37.1 | 0.453 | 4.17e-26 | 5.96e-23 |
| BBS4 | 15q24.1 | 0.443 | 6.34e-25 | 7.93e-22 |
| LGALSL | 2p14 | 0.440 | 1.59e-24 | 1.87e-21 |
| ZNF582 | 19q13.43 | 0.438 | 2.67e-24 | 2.97e-21 |
| TUT7 | 9q21.33 | 0.438 | 2.86e-24 | 3.00e-21 |
| DCAF6 | 1q24.2 | 0.436 | 4.61e-24 | 4.61e-21 |
| FAM13C | 10q21.1 | 0.436 | 4.90e-24 | 4.67e-21 |
| STEAP2 | 7q21.13 | 0.435 | 5.93e-24 | 5.39e-21 |
| STEAP1 | 7q21.13 | 0.432 | 1.28e-23 | 1.11e-20 |
| REPS2 | Xp22.2 | 0.430 | 2.01e-23 | 1.60e-20 |
| ZFYVE9 | 1p32.3 | 0.430 | 2.47e-23 | 1.90e-20 |
| LIG3 | 17q12 | 0.427 | 4.99e-23 | 3.60e-20 |
| CAMKK2 | 12q24.31 | 0.427 | 5.05e-23 | 3.60e-20 |
| NBEAP1 | 15q11.2 | 0.426 | 5.80e-23 | 4.00e-20 |
| DMXL1 | 5q23.1 | 0.426 | 6.52e-23 | 4.34e-20 |
| DOCK7 | 1p31.3 | 0.425 | 7.58e-23 | 4.89e-20 |
| KLK3 | 19q13.33 | 0.425 | 8.73e-23 | 5.46e-20 |
| ALG6 | 1p31.3 | 0.420 | 2.50e-22 | 1.47e-19 |
| ARHGEF26 | 3q25.2 | 0.418 | 4.59e-22 | 2.62e-19 |
| SMS | Xp22.11 | 0.418 | 4.85e-22 | 2.69e-19 |
| PLPP6 | 9p24.1 | 0.417 | 5.50e-22 | 2.95e-19 |
| NUDT4 | 12q22 | 0.417 | 5.60e-22 | 2.95e-19 |
| SUOX | 12q13.2 | 0.415 | 9.57e-22 | 4.85e-19 |
| STEAP1B | 7p15.3 | 0.415 | 9.70e-22 | 4.85e-19 |
| HERC3 | 4q22.1 | 0.413 | 1.44e-21 | 7.01e-19 |
| MAP2 | 2q34 | 0.413 | 1.51e-21 | 7.20e-19 |
| SCOC | 4q31.1 | 0.413 | 1.57e-21 | 7.29e-19 |
| OSBP | 11q12.1 | 0.413 | 1.71e-21 | 7.76e-19 |
| TRIM68 | 11p15.4 | 0.411 | 2.85e-21 | 1.24e-18 |
| SLC39A6 | 18q12.2 | 0.409 | 3.73e-21 | 1.55e-18 |
| SLC36A1 | 5q33.1 | 0.409 | 4.25e-21 | 1.73e-18 |
| ACACA | 17q12 | 0.408 | 5.12e-21 | 2.05e-18 |

E

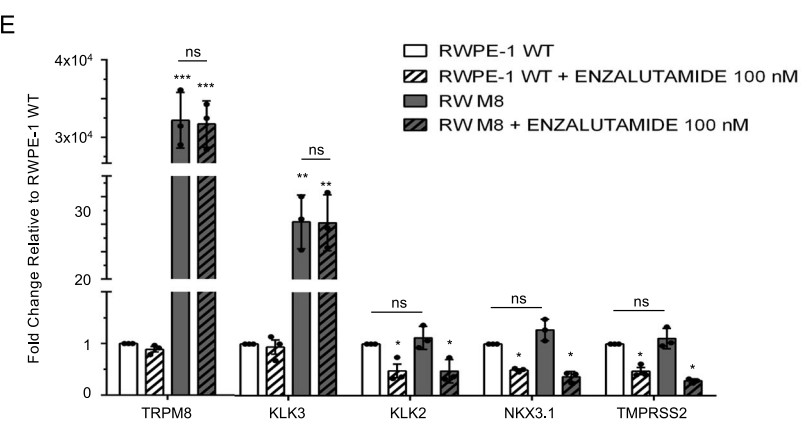

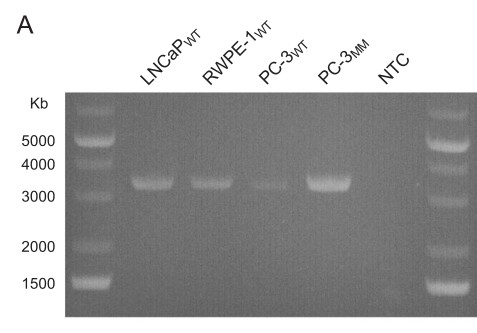

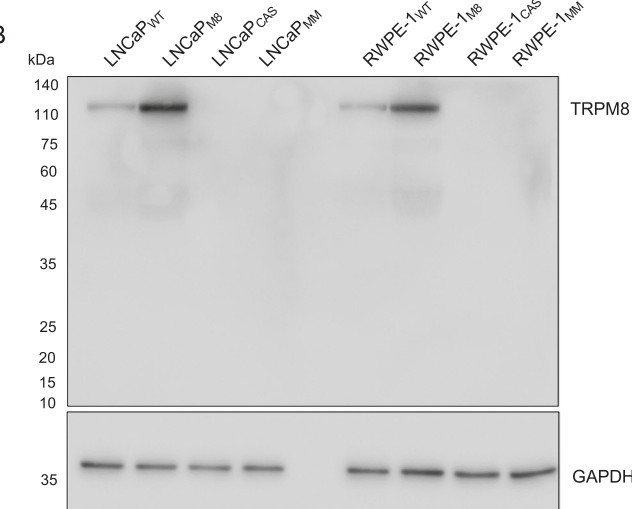

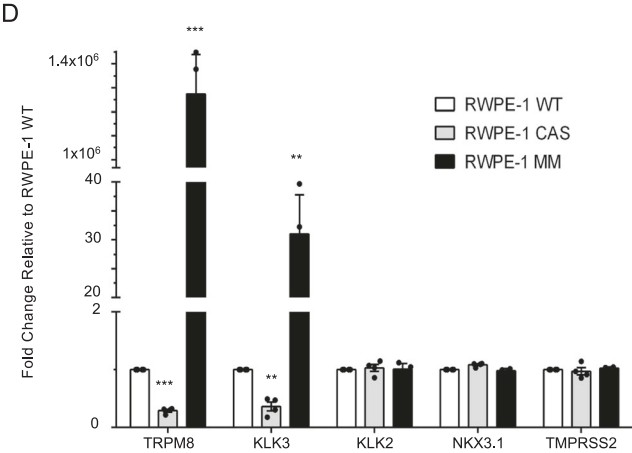

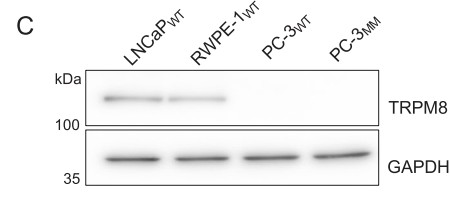

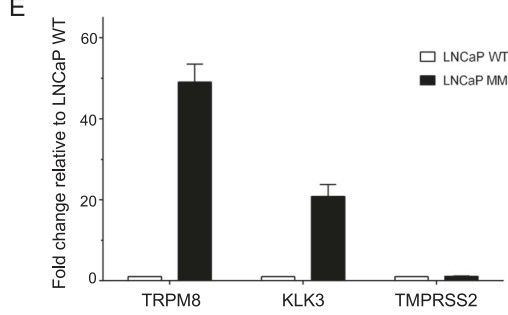

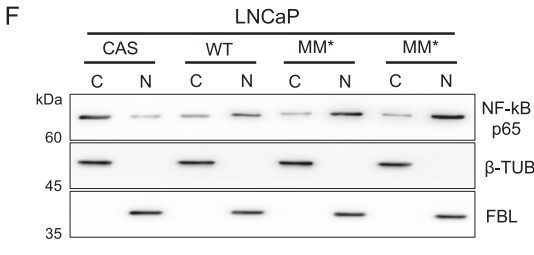

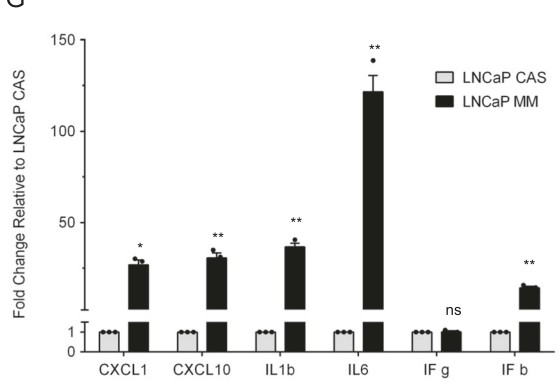

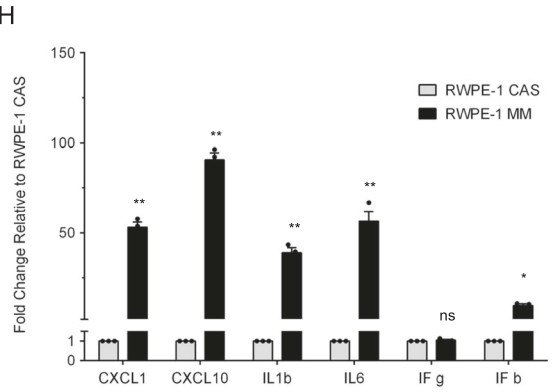

**Figure EV2.** **Pro-inflammatory activity of the mutant non-coding RNA of TRPM8.**

(A,B) End-point PCR (A) and Western blot analysis (B) of TRPM8 in LNCaP (WT), RWPE-1 (WT) and PC-3 (WT and MM) cell lines. GAPDH was used as loading control. (C) Immunoblotting analysis of TRPM8 protein with the Alomone (#ACC-049) antibody in wild type and genetically modified (M8, CAS, MM) LNCaP and RWPE-1 cell lines. (D) RT-qPCR analysis of *TRPM8, KLK3* and *TMPRSS2* genes in both wild type (WT) and genetically engineered (MM, two independent clones) LNCaP cell lines expressing reduced levels of TRPM8 RNA compared to clones described in Fig. 1C. (E) Immunoblot analysis of NF-kB p65 in cytosolic and nuclear fractions of CAS, WT, and MM (described in D) engineered LNCaP cell lines. β-Tubulin (β-TUB) and Fibrillarin (FBL) were used as cytoplasmic and nuclear markers, respectively. (F,G) RT-qPCR analyses of canonical NF-kB and IRF3 targets genes in both LNCaP and RWPE-1 CAS and MM cells. Data Information: In (D,E,G,H) data are presented as mean ± SD of $n = 4$ (D), $n = 2$ (E), and $n = 3$ (G,H) independent biological replicates. *$P ≤ 0.05$; **$P ≤ 0.01$; ns, not statistically significant (Two-tailed Student's *t*-test). Source data are available online for this figure.

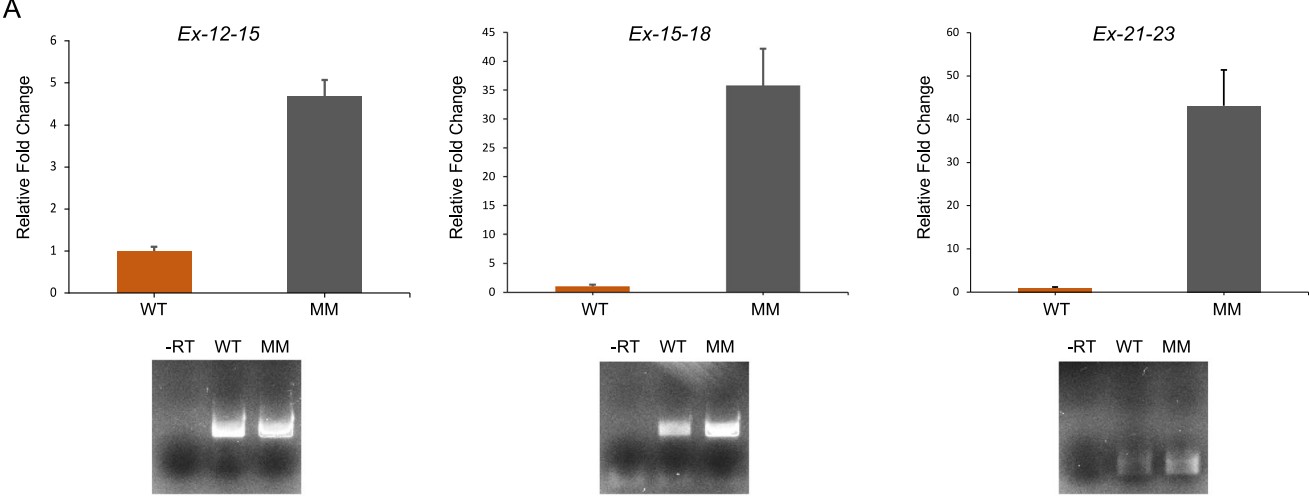

**A**

*Ex-12-15*

>WT
CTCCAACCACTTCAGCACGCTTGTGTACCGGAATCTGCAGAT
CGCCAAGAATTCCTATAATGATGCCCTCCTCACGTTTGTCTG
GAAACTGGTTGCGAACTTCCGAAGAGGCTTCCGGAAGGAAG
ACAGAAATGGCCGGGACGAGATGGACATAGAACTCCACGAC
GTGTCTCCTATTACTCGGCACCCCCTGCAAGCTCTCTTCATC
TGGGCCATTCTTCAGAATAAGAAGGAACTCTCCAAAGTCATT
TGGGAGCAGACCAGGGGCTGCACTCTGGCAGCCCTGGGAG
CCAGCAAGCTTCTGAAGACTCTGGCCAAAGTGAAGAA
>MM
CTCCAACCACTTCAGCACGCTTGTGTACCGGAATCTGCAGA
TCGCCAAGAATTCCTATAATGATGCCCTCCTCACGTTTGTCT
GGAAACTGGTTGCGAACTTCCGAAGAGGCTTCCGGAAGGAA
GACAGAAATGGCCGGGACGAGATGGACATAGAACTCCACGA
CGTGTCTCCTATTACTCGGCACCCCCTGCAAGCTCTCTTCAT
CTGGGCCATTCTTCAGAATAAGAAGGAACTCTCCAAAGTCAT
TTGGGAGCAGACCAGGGGCTGCACTCTGGCAGCCCTGGGA
GCCAGCAAGCTTCTGAAGACT

*Ex-15-18*

>WT
GATTATCCTGTGTCTGTTTATTATACCCTCGGTGGGCTGTGGC
TTTGTATCATTTAGGAAGAAACCTGTCGACAAGCACAAGAAGC
TGCTTTGGTACTATGTGGCGTTCTTCACCTCCCCCTTCGTGG
TCTTCTCCTGGAATGTGGTCTTCTACATCGCCTTCCTCCTGCT
GTTTGCCTACGTGCTGCTCATGGATTTCCATTCGGTGCCACA
CCCCCCGAGCTGGTCCTGTACTCGCTGGTCTTTGTCCTCTT
CTGTGATGAAGTGAGACAGTGGTACGTAAATGGGGTGAATTA
TTTTACTGACCTGTGGAATGTGA
>MM
GATTATCCTGTGTCTGTTTATTATACCCTTGGTGGGCTGTGGC
TTTGTATCATTTAGGAAGAAACCTGTCGACAAGCACAAGAAGC
TGCTTTGGTACTATGTGGCGTTCTTCACCTCCCCCTTCGTGGT
CTTCTCCTGGAATGTGGTCTTCTACATCGCCTTCCTCCTGCTG
TTTGCCTACGTGCTGCTCATGGATTTCCATTCGGTGCCACAC
CCCCCGAGCTGGTCCTGTACTCGCTGGTCTTTGTCCTCTTC
TGTGATGAAGT

*Ex-21-23*

>WT
TGCATCTACTTGTTATCCACCAACATCCAGCTGGTCAACCAG
CTGGTCGCCATGTTTGGCTACACGGTGGGCACCGTCCAGG
AGAACAATGACCAGGTCTGGAAGTTCCAGAGGTACTTCCTG
GTGCAGGAGTACTGCAGCCGCCTCAATATCCCCTTCCCCTT
CATCGTCTTCGCTTACTTCTACATGGTGGTGAAGAAGTGCTT
CAAGTGTTGCTGCAAGGAGAAAAACATGGAGTCTTCTGTCT
GCTGTTTCAAAAATGAAGACAATGAGACTCTGGC
>MM
TGCATCTACTTGTTATCCACCAACATCCTGCTGGTCAACCTG
CTGGTCGCCATGTTTGGCTACACGGTGGGCACCGTCCAGGA
GAACAATGACCAGGTCTGGAAGTTCCAGAGGTACTTCCTGG
TGCAGGAGTACTGCAGCCGCCTCAATATCCCCTTCCCCTTC
ATCGTCTTCGCTTACTTCTACATGGTGGTGAAGAAGTGCTTC
AAGTGTTGCTGCAAGGAGAAAAACATGGAGTCTTCTGTCTGC
TGTTTCAAAAATGAAGACAATGAGACTCTGGCATGGGAGGGT
GTCCAATGAAAGGAA

**B**

*Supernatant LNCaP WT*

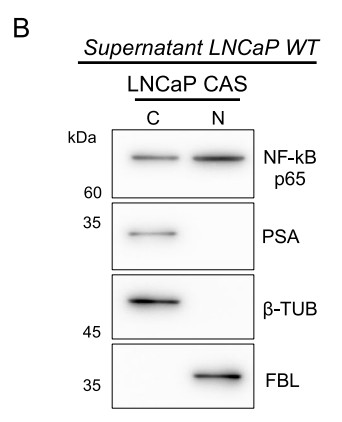

**C**

*RWPE-1 Supernatant*

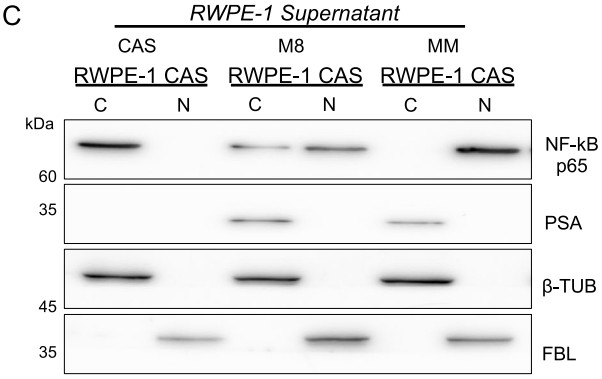

**D**

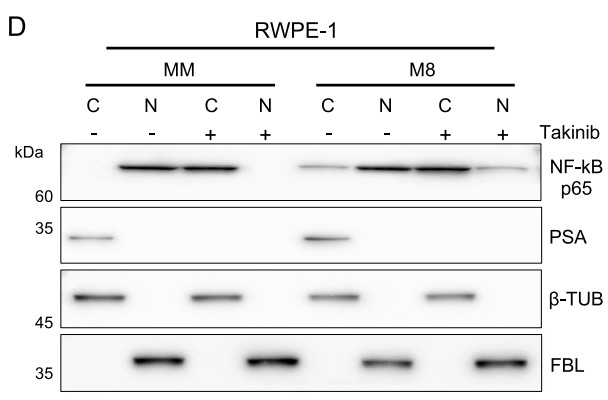

**E**

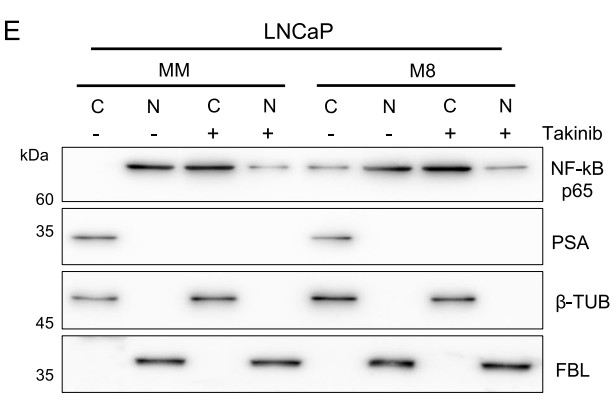

◀ **Figure EV3.   TRPM8 RNA secretion modulates TLR3/TAK1 signal cascade.**

(**A**) RT-qPCR analysis of TRPM8 transcript released by WT and MM LNCaP cells into extracellular vesicles (EVs). Specificity and sequence of amplicons obtained with sets of primers spanning different exons (Ex) of the coding sequence of TRPM8 RNA are shown. *Y3* transcript was used to normalize the data. (**B**) Nucleus/Cytosol fractionation and Western blot analysis of NF-kB p65 and PSA in LNCaP CAS cells conditioned with the supernatants of LNCaP WT cells. β-Tubulin (β-TUB) and Fibrillarin (FBL) were used as markers of the cytosolic and nuclear fractions, respectively. (**C**) Nucleus/Cytosol fractionation and Western blot analysis of NF-kB p65 and PSA in RWPE-1 CAS cells conditioned with the supernatants of RWPE-1 (CAS, M8, and MM). β-Tubulin (β-TUB) and Fibrillarin (FBL) were used as markers of the cytosolic and nuclear fractions, respectively. (**D,E**) Nucleus/Cytosol fractionation and Western blot analysis of NF-kB p65 and PSA in RWPE-1 MM and M8 (**D**) and LNCaP MM and M8 (**E**) cell lines in the presence or absence of Takinib (TAK1 inhibitor; 10 μM, 24 h). β-Tubulin (β-TUB) and Fibrillarin (FBL) were used as markers of the cytosolic and nuclear fractions, respectively. Data Information: In (**A**) data are presented as mean ± SD of $n = 3$ biological replicates. Source data are available online for this figure.

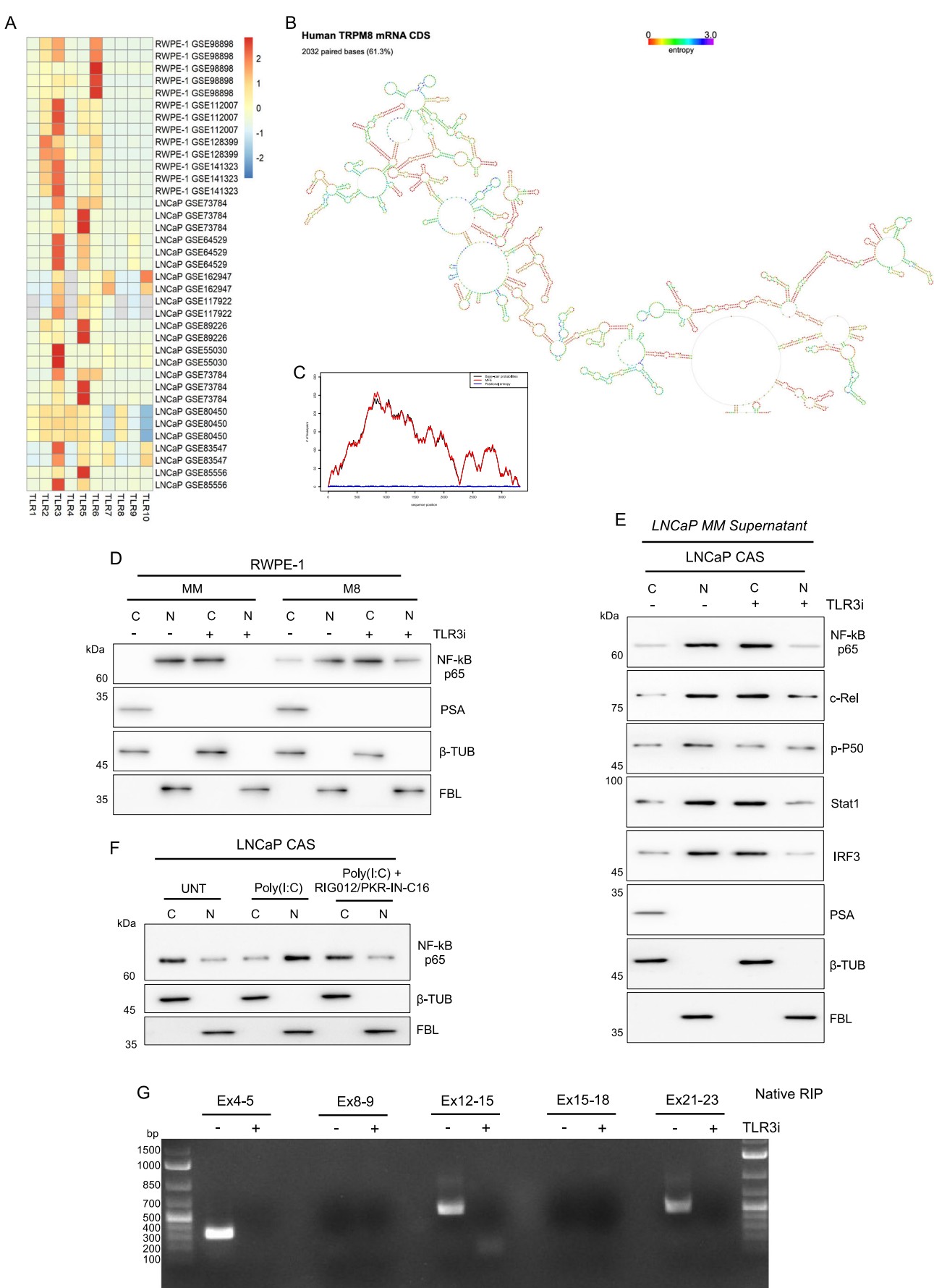

◄ **Figure EV4. Prediction of TRPM8 RNA structures highlights putative TLR3 binding motifs.**

(A) TLRs expression in LNCaP and RWPE-1 prostate cell lines based on publicly available datasets. (B) In silico prediction of the secondary structure of human TRPM8 transcript. Highly stable dsRNA stems with bulge/internal loops are indicated in red. (C) Mountain plot of (B). (D) Nucleus/Cytosol fractionation and Western blot analysis of NF-kB p65 and PSA in RWPE-1 MM and M8 cell lines in the presence or absence of TLR3/dsRNA Complex Inhibitor (TLR3 inhibitor; 20 µM, 24 h). β-Tubulin (β-TUB) and Fibrillarin (FBL) were used as markers of the cytosolic and nuclear fractions, respectively. (E) Nucleus/Cytosol fractionation and Western blot analysis of NF-kB p65, c-Rel, phospho-p50, STAT1, IRF3 and PSA in LNCaP CAS cells conditioned with the supernatants of LNCaP MM cells. β-Tubulin (β-TUB) and Fibrillarin (FBL) were used as markers of the cytosolic and nuclear fractions, respectively. (F) Nucleus/Cytosol fractionation and Western blot analysis of NF-kB p65 in LNCaP CAS cells transfected with Poly(I:C) (2 µg/mL, 24 h) and treated or not with the the combination of the RIG1 inhibitor RIG012 (2 µM, 6 h) and the PKR inhibitor C16 (2 µM, 6 h). Untransfected and untreated LNCaP CAS cells served as control for the activation of NF-kB by Poly(I:C). β-Tubulin (β-TUB) and Fibrillarin (FBL) were used as markers of the cytosolic and nuclear fractions, respectively. (G) PCR analysis of TLR3-bound TRPM8 transcript (native RIP) in LNCaP WT cells treated or not with TLR3/dsRNA complex inhibitor. Sets of primers spanning different exons (Ex) of the coding sequence of TRPM8 RNA are shown. Data Information: In (A) the heatmap was plotted with the pheatmap package, scaling gene expression by row. All the analyses were done with R 4.0.3. Source data are available online for this figure.

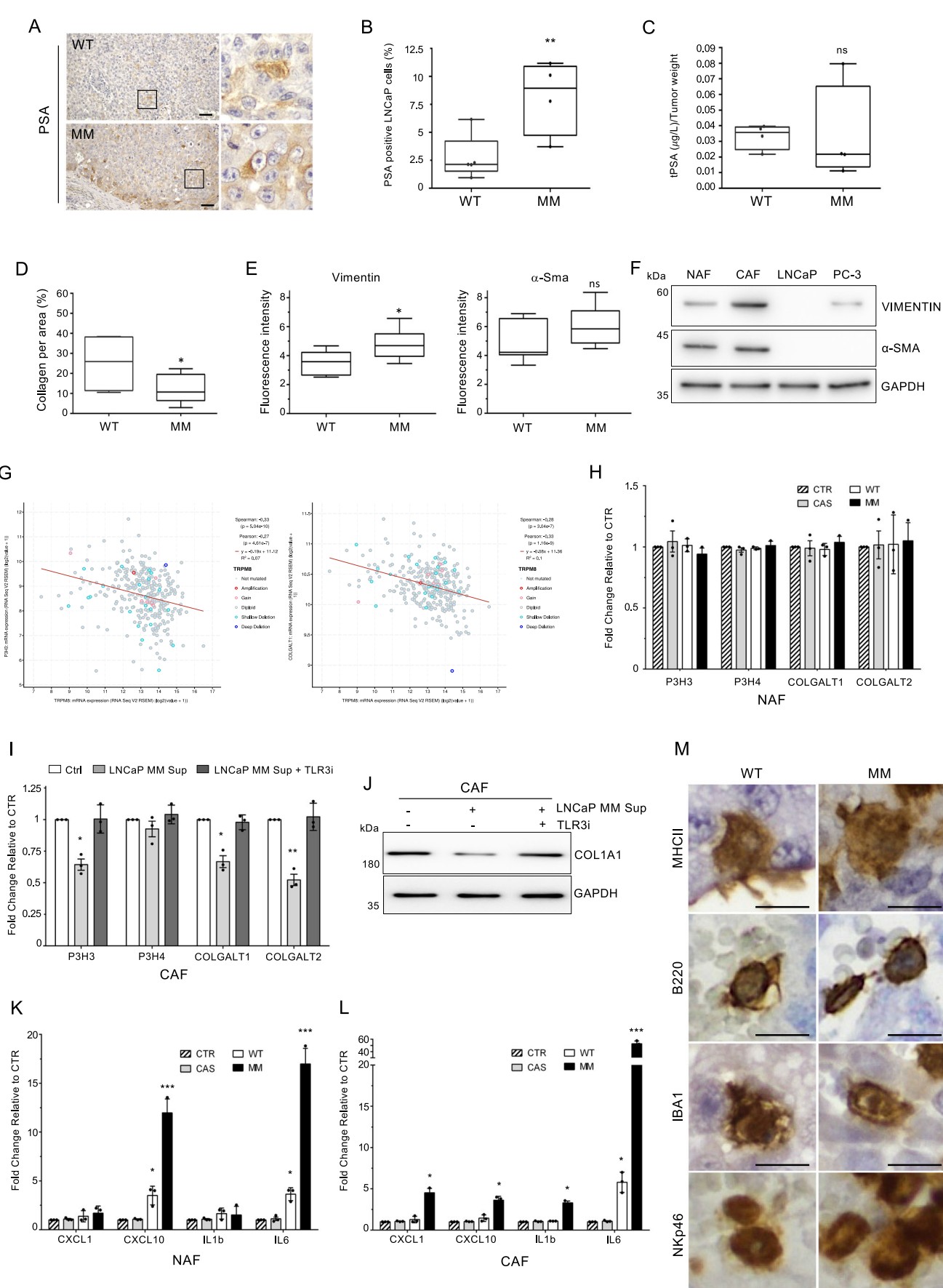

**Figure EV5. TRPM8 RNA secretion alters fibroblasts functions.**

(A,B) Immunolocalization of PSA in FFPE sections of LNCaP WT ($n = 3$ xenografts/5 sections) and MM ($n = 2$ xenografts/4 sections) tumors (A), coupled with the percentage of PSA+ cells (total number of cells counted: WT = 142.293; MM = 106.013) (B). (C) Quantification of total PSA (free and complexed) circulating in the plasma of xenografted mice (WT, $n = 4$; MM, $n = 4$). (D) Quantification of Collagen Type-I (percentage of blue area to total section area; $n = 6$ sections per genotype), relative to Fig. 4D. (E) Quantification of Vimentin and α-Sma in FFPE sections of LNCaP WT and LNCaP MM xenograft (a minimum of 5 different areas per genotype were analyzed), relative to Fig. 4E. (F) Western blot analysis of Vimentin and α-SMA proteins in NAF, CAF, LNCaP and PC-3 cell lines. GAPDH was used as loading control. (G) Inverse correlation in PCa of the expression of *P3H3* (left panel) and *COLGALT1* (right panel) with respect to *TRPM8* (TCGA, cBioportal). (H) RT-qPCR analysis of the indicated genes of the metabolism of type I collagen in NAF cells. (I) RT-qPCR analysis of *PH3, PH4, COLGALT1*, and *COLGALT2* genes expression in CAFs conditioned with the supernatant of LNCaP MM cells and treated or not with the TLR3 inhibitor (20 μM, 24 h). (J) Western blot analysis of COL1A1 levels in CAFs conditioned with the supernatant of LNCaP MM cells and treated or not with the TLR3 inhibitor (XX μM, 24 h). GAPDH was used as loading control. (K,L) RT-qPCR analysis of canonical NF-kB targets genes in NAF (K) and CAF (L) cells conditioned with the supernatants of LNCaP (CAS, WT, MM) cell lines. (M) Magnification images of MHCII, CD45R (B220), IBA1 and NKp46 immunolocalization in FFPE sections of LNCaP wild type (WT) and LNCaP MM xenografts. Scale bars, 20 μm. Data Information: In (B–E) box-plots elements indicate the median (center line), upper and lower quartiles (box limits). Whiskers extend from the minimum to the maximum. *$P \leq 0.05$; **$P \leq 0.01$. (Student's *t*-test). In (H,I,K,L) data are presented as mean ± SD of $n = 3$ independent biological replicates. *$P \leq 0.05$; **$P \leq 0.01$; ***$P \leq 0.001$. (Two-tailed Student's *t*-test). Source data are available online for this figure.

