## [Peer Review File · The EMBO Journal]

Sterile inflammation via TRPM8 RNA-dependent TLR3-NF- κ B/IRF3 activation promotes antitumor immunity in prostate cancer

Alessandro Alaimo, Sacha Genovesi, Nicole Annesi, Dario De Felice, Saurav Subedi, Alice Macchia, Federico La Manna, Yari Ciani, Federico Vannuccini, Vera Mugoni, Michela Notarangelo, Michela Libergoli, Francesca Broso, Riccardo Taulli, Ugo Ala, Aurora Savino, Martina Cortese, Somayeh Mirzaaghaei, Valeria Poli, Ian-Marc Bonapace, Mauro Papotti, Luca Molinaro, Claudio Doglioni, Orazio Caffo, Adriano Anesi, Michael Nagler, Giovanni Bertalot, Francesco Carbone, Mattia Barbareschi, Umberto Basso, Erik Dassi, Massimo Pizzato, Alessandro Romanel, Francesca Demichelis, Marianna Kruithof-de Julio, and Andrea Lunardi

Corresponding author: Andrea Lunardi (andrea.lunardi@unitn.it)

Review Timeline:

Submission Date:	11th May 23
Editorial Decision:	28th Jul 23
Revision Received:	17th Nov 23
Editorial Decision:	22nd Dec 23
Revision Received:	6th Jan 24
Accepted:	10th Jan 24

Editor: Daniel Klimmeck

Transaction Report:

Dear Dr Lunardi,

Thank you again for the submission of your manuscript (EMBOJ-2023-114493) to The EMBO Journal, and providing us with a preliminary point-by-point response to the concerns raised by the referees. Please accept my sincere apologies for getting back to you with unusual protraction due to delayed referee input, as well as detailed discussion in the editorial team. As mentioned, your study was assessed by two reviewers with expertise in cancer biology, whose comments are enclosed below.

As you will see from their comments, the referees acknowledge the analysis and potential interest and value of your findings. However, they also express major concerns i.p. regarding the endogenous relevance of the proposed axis, pathophysiological importance of your results and direct causalities of the involved factors.

Given the overall interest stated and broader angle of your results, we are able to invite you to revise your manuscript experimentally to address the referees' comments, along the lines sketched in your outline. I need to stress though that we do require strong support from the referees on a revised version of the study in order to move on to publication of the work. Specifically, referee comments will be important to judge the relevance of additional experimentation proving TRPM8 transcript-TLR3 interaction (ref#2, pt.5). As to the open outcome of the revisional work I suggest keeping EMBO Reports in mind for this study as an alternative venue.

Please feel free to contact me if you have any questions or need further input on the referee comments.

When submitting your revised manuscript, please carefully review the instructions below.

Please feel free to approach me any time should you have additional questions related to this.

Thank you for the opportunity to consider your work for publication.

I look forward to your revision.

Kind regards,

Daniel Klimmeck

Daniel Klimmeck, PhD
Senior Editor
The EMBO Journal

Instruction for the preparation of your revised manuscript:

- 1) a .docx formatted version of the manuscript text (including legends for main figures, EV figures and tables). Please make sure that the changes are highlighted to be clearly visible.
- 2) individual production quality figure files as .eps, .tif, .jpg (one file per figure).
- 3) a .docx formatted letter INCLUDING the reviewers' reports and your detailed point-by-point response to their comments. As part of the EMBO Press transparent editorial process, the point-by-point response is part of the Review Process File (RPF), which will be published alongside your paper.
- 4) a complete author checklist, which you can download from our author guidelines ([https://wol-prod-cdn.literatumonline.com/pb-assets/embo-site/Author Checklist%20-%20EMBO%20J-1561436015657.xlsx](https://wol-prod-cdn.literatumonline.com/pb-assets/embo-site/Author%20Checklist%20-%20EMBO%20J-1561436015657.xlsx)). Please insert information in the checklist that is also reflected in the manuscript. The completed author checklist will also be part of the RPF.
- 5) Please note that all corresponding authors are required to supply an ORCID ID for their name upon submission of a revised

manuscript.

6) It is mandatory to include a 'Data Availability' section after the Materials and Methods. Before submitting your revision, primary datasets produced in this study need to be deposited in an appropriate public database, and the accession numbers and database listed under 'Data Availability'. Please remember to provide a reviewer password if the datasets are not yet public (see <https://www.embopress.org/page/journal/14602075/authorguide#datadeposition>).

7) Our journal encourages inclusion of *data citations in the reference list* to directly cite datasets that were re-used and obtained from public databases. Data citations in the article text are distinct from normal bibliographical citations and should directly link to the database records from which the data can be accessed. In the main text, data citations are formatted as follows: "Data ref: Smith et al, 2001" or "Data ref: NCBI Sequence Read Archive PRJNA342805, 2017". In the Reference list, data citations must be labeled with "[DATASET]". A data reference must provide the database name, accession number/identifiers and a resolvable link to the landing page from which the data can be accessed at the end of the reference. Further instructions are available at .

8) At EMBO Press we ask authors to provide source data for the main and EV figures. Our source data coordinator will contact you to discuss which figure panels we would need source data for and will also provide you with helpful tips on how to upload and organize the files.

Numerical data can be provided as individual .xls or .csv files (including a tab describing the data). For 'blots' or microscopy, uncropped images should be submitted (using a zip archive or a single pdf per main figure if multiple images need to be supplied for one panel). Additional information on source data and instruction on how to label the files are available at .

9) We replaced Supplementary Information with Expanded View (EV) Figures and Tables that are collapsible/expandable online (see examples in <https://www.embopress.org/doi/10.15252/embj.201695874>). A maximum of 5 EV Figures can be typeset. EV Figures should be cited as 'Figure EV1, Figure EV2' etc. in the text and their respective legends should be included in the main text after the legends of regular figures.

11) For data quantification: please specify the name of the statistical test used to generate error bars and P values, the number (n) of independent experiments (specify technical or biological replicates) underlying each data point and the test used to calculate p-values in each figure legend. The figure legends should contain a basic description of n, P and the test applied. Graphs must include a description of the bars and the error bars (s.d., s.e.m.).

The revision must be submitted online within 90 days; please click on the link below to submit the revision online before 26th Oct 2023.

Link Not Available

Referee #1:

In the manuscript by Alaimo et al., the authors uncover and investigate mechanistically a new role for TRPM8 RNA in sterile inflammation and its potential connection to prostate cancer development. TRPM8 is a cation channel expressed in luminal cells of the human normal prostate which exhibits elevated levels in prostate cancer. Activation of TRMP8 through specific agonists and subsequent prolonged increase of calcium flux has been previously shown by the same group to activate cell death and sensitize prostate cancer cells to radio, chemo or hormonal therapy. In the current study, the authors focus on the role of TRPM8 RNA released from prostate cancer cells into exosomes and demonstrate that it might trigger an inflammatory response in tumor cells and tumor microenvironment. The authors use overexpression (M8) and downregulation (Crispr- Cas or siRNA) of TRPM8 as well as overexpression of a translation-defective TRPM8 RNA (MM) in LNCaP hormone-sensitive metastatic prostate cancer cells and immortalized normal prostate cells RWPE-1 and xenografts to examine the impact of TRPM8 RNA on tumor formation. The authors pursue a novel link with therapeutic potential and so far, there is no other study to explore the role of exosomal TRPM8 RNA in prostate inflammation and cancer. However, there are a number of concerns regarding the experimental approaches to uncover the TRPM8 RNA function and overstatement of some conclusions based on the provided data.

Major critiques:

- 1) The manuscript relies on overexpression in cell lines of TRPM8 (M8) and of a mutated TRPM8 cDNA that is translation-defective and does not generate a functional TRPM8 protein and thus allows investigation of the TRPM8 RNA role. These appear to be expressed at very high levels and it is unclear how they correlate with in vivo intra-tumor levels or whether TRPM8 wt RNA levels found in vitro can elicit the same induction of Nf-kB pathway activation as these very high levels. In which conditions in the tumor such high levels would be observed? More data from human tumors and a more thorough investigation of the in vivo xenografts would likely strengthen this point.
- 2) The manuscript in general would benefit from a more clear hypothesis and some re-organization to provide a more streamlined connection between TRPM8-Tlr3-Nf-Kb. The implications of increased PSA upon TRPM8 RNA increase in exosomes should also be stated more clearly. Both TAK and TLR3 inhibitors could have effects on the NFKB pathway independent of TRMP8 or in response to other components of the exosomes. A direct connection of TRPM8 RNA with TLR3 does not appear to have been established. Supernatant experiments could be performed with TLR3i to strengthen this point. What other components of the NF-Kb pathway are activated? In addition, NF-KB is part of numerous pathways, are other inflammatory pathways activated?
- 3) A lack of effect of MM translation-defective mutant on proliferation and cell death is claimed, however, only cell cycle analyses by FACS are shown. Images of cell cultures for morphology, growth curves and some cell death assays should be performed to support this conclusion. The authors performed similar assays in a previous publication where they characterized the effects of the M8 overexpression, but the MM mutant has not been previously characterized.
- 4) Major conclusions are drawn based on the MM mutant. Does this RNA have the same conformation as the WT RNA in terms of secondary structure? Could the mutant be more immunogenic? If this is the case, this should be addressed.
- 5) Trichrome staining in Fig. 4D should be supported by quantification, same for SMA and vimentin in Fig. 4E, especially since vimentin levels appear increased in the WB of MM xenografts in Fig. 4C.
- 6) While the LNCap cell lines might generate EVs containing TRPM8 RNA, the TRPM8 RNA in blood exosomal vesicles might not be derived from cancer cells. Can the source/origin of blood exosomal TRPM8 be addressed? Is it increased in tumor cells from the same patients as well?

Minor critiques:

- 7) The abstract text would benefit from some rewording to increase clarity especially in regards to the connection of the sterile inflammation to cancer.
- 8) Expanded Figure 1D (Fig. S1D): is PSA WB from the same probing? If this is from a different WB, this should be stated.
- 9) Fig. 2E: since RWPE1 cells do not express detectable Nkx3-1 levels, another unchanged marker should be shown.
- 10) Expanded Figure 2: PC3 cells are shown, but not mentioned in the main text.
- 11) Fig. 3D, 3E: WT supernatant is mentioned in the text, however only supernatant from CAS, M8, and MM are shown. Could

this be clarified? Expanded Figure 3A shows the LNCap WT supernatant results, but not the WT RWPE1 supernatant.

12) Could the NF- κ B p65 nuclear shuttling be supported by imaging? Are all cells responding in the same way?

13) Fig 4C: a loading control appears to not have been included.

14) Is there a hypothesis behind including NAFs and CAFs? This should be made more clear in the text. What is the explanation for an effect of MM supernatant on CAFs, but not on NAFs in Fig. 4H? The changes in CAFs are more consistent, but some (such as CXCL10) appear modest in CAFs compared to NAFs. Is there a conclusion/suggestion to be made of these results?

15) How are NF- κ B and TLR3 affected in MM or ME xenografts?

16) While the tumors appear histologically different, there seems to be no change in tumor burden. Could some treatments be included in this study to strengthen the clinical implications?

17) TRPM8 has been shown to exert an inhibitory effect on cell migration. Does TRPM8 RNA play a role in cell migration?

18) In the de Souza reference included in the text, although TRPM8 RNA is present in blood, there is no significant difference by qPCR between TRPM8 RNA in prostate cancer patients blood compared to normal blood. How are the blood EVs results from this manuscript related to the de Souza results? Are blood EVs enriched for TRPM8 RNA in prostate cancer patients?

Minor suggestions:

The authors should check for few typos, there are also few paragraphs missing some words or having some extra words. For instance, the last sentences on pg. 7 need some re-writing.

Referee #2:

In the manuscript entitled, "Androgen-independent PSA spikes unmask a novel mechanism of sterile inflammation with critical roles in prostate tumorigenesis," the authors outlined a novel inflammatory mechanism in the prostate epithelium caused by the interaction of TRMP8 RNA with TLR3. They show that TRMP8 regulates KLK3 (PSA) expression in an AR-independent mechanism in both immortalized and prostate cancer cell lines. They demonstrate that enforced expression of TRMP8 RNA regulates NF- κ B activation and PSA induction without interfering with cell proliferation and survival. Furthermore, they found that TRPM8 RNA contributes to NF- κ B activation through endosomal TLR3 signaling through translational-independent mechanisms within extracellular vesicles (EVs). As a result of TRPM8-induced inflammation in PCa xenografts, NK cells infiltrated more readily, and cancer-associated fibroblasts (CAFs) decreased collagen type I deposition, likely contributing to anti-tumor immunity. While the mechanism is quite interesting, the heavy dependence on an overexpression system raises the important question of physiologic relevance. We point out specific areas in this manuscript that the authors can consider to improve this manuscript:

Major Comments:

1) The authors claim that "Unexpectedly, the results were recapitulated even under complete androgen blockade in Fig 1D." However, KLK3 and other AR transcriptional targets are not affected by LNCaP WT Enzalutamide treatment. This is a very peculiar findings and goes against published literature.

2) In Fig 1H-I, the authors conclude that "the PSA amount in both LNCaP and RWPE-1 cells was not influenced by gating TRPM8 channel with its potent agonist WS-12 or antagonist AMBT (Fig 1H and I), suggesting that channel activity is not involved in the observed effects associated with TRPM8 overexpression." Since TRPM8 is a Ca²⁺ uptake channel, the authors should quantify the Ca²⁺ efflux response to conclude no involvement of TRPM8 channel activity in PSA regulation. This can be efficiently done using fluorescent calcium indicators like Fura-2 AM and Fluo-4.

3) In Fig 3D and E, the authors show that supernatants of WT, M8, and MM containing TRMP8 RNA activates NF κ B signaling. However, they do not show cytosolic and nuclear NF- κ B p65 levels in LNCaP and RWPE-1 cells without supernatant.

4) Also why did the authors use supernatant in Fig 3D and E instead of purified EVs? This is a major confounding factor of this experiment because supernatant is rich with many other things besides EVs.

5) In Fig EV3C and D, the authors show in silico prediction of TRPM8 RNA structure with stable stems and loops. However, this does not confirm that these structures of TRPM8 RNA binds to TLR3. Since this is one of the key conclusions of the proposed model, this needs to be proven experimentally. The authors could pull down the TLR3/ligand complex and RT for TRMP8 RNA.

6) In Fig 4F authors show that the CAFs reduces COL1A1 protein when conditioned with MM supernatant. Is this effect only due

to TRPM8 MM RNA? Can this be an indirect effect because of other secretory differences besides having more TRMP8 RNA?

7) The authors confirmed that LNCaP MM xenografts show a higher amount of PSA in the tissue however the circulating total (free + complexed) PSA was the opposite. What does this mean and how does it fit the proposed model?

Minor comments

1. On page 5 authors say, "Once stably overexpressed in both TRPM8 knocked-out (CAS) LNCaP and RWPE-1 cell lines, the TRPM8 mutant transcript (MM) was shown to be stable (Fig 2B) but unable to encode for the protein (Fig EV2A-C)." This sentence needs restructuring.
2. The schematic of potential mechanism of TRPM8 RNA/TLR3 molecular circuit in Fig 3C should be at the end of Fig 3. So that the authors have shown the TLR3 NFKB activation related experiments before presenting the model.
3. In general, these figures and their legends need additional labeling and experimental details: Fig EV2D, Fig EV2E, Fig 3A, Fig 3B, Fig 4B, Fig 4D.

Manuscript #EMBOJ-2023-114493

We are honestly grateful to our Reviewers for their insightful comments and suggestions that substantially improved the quality and robustness of our findings and strengthened our conclusions. All the points raised by the Reviewers have been successfully addressed as carefully described in the point-by-point rebuttal below. All changes introduced to the manuscript and figure legends are highlighted in yellow. We hope that the manuscript will now be considered suitable for publication in EMBO Journal.

Point by point rebuttal

Referee #1:

In the manuscript by Alaimo et al., the authors uncover and investigate mechanistically a new role for TRPM8 RNA in sterile inflammation and its potential connection to prostate cancer development. TRPM8 is a cation channel expressed in luminal cells of the human normal prostate which exhibits elevated levels in prostate cancer. Activation of TRMP8 through specific agonists and subsequent prolonged increase of calcium flux has been previously shown by the same group to activate cell death and sensitize prostate cancer cells to radio, chemo or hormonal therapy. In the current study, the authors focus on the role of TRPM8 RNA released from prostate cancer cells into exosomes and demonstrate that it might trigger an inflammatory response in tumor cells and tumor microenvironment. The authors use overexpression (M8) and downregulation (Crispr- Cas or siRNA) of TRPM8 as well as overexpression of a translation-defective TRPM8 RNA (MM) in LNCaP hormone-sensitive metastatic prostate cancer cells and immortalized normal prostate cells RWPE-1 and xenografts to examine the impact of TRPM8 RNA on tumor formation. The authors pursue a novel link with therapeutic potential and so far, there is no other study to explore the role of exosomal TRPM8 RNA in prostate inflammation and cancer.

However, there are a number of concerns regarding the experimental approaches to uncover the TRPM8 RNA function and overstatement of some conclusions based on the provided data.

We thank the Reviewer to underline the novelty of this work and its clinical value which stems from the large cohort of urologists, oncologists, and pathologists that co-authored this work. We capitalized on the relevant criticisms raised by the Reviewer and strengthened our findings and conclusions.

Major critiques:

1) The manuscript relies on overexpression in cell lines of TRPM8 (M8) and of a mutated TRPM8 cDNA that is translation-defective and does not generate a functional TRPM8 protein and thus allows investigation of the TRPM8 RNA role.

These appear to be expressed at very high levels and it is unclear how they correlate with in vivo intra-tumor levels or whether TRPM8 wt RNA levels found in vitro can elicit the same induction of Nf-kB pathway activation as these very high levels. In which conditions in the tumor such high levels would be observed? More data from human tumors and a more thorough investigation of the in vivo xenografts would likely strengthen this point.

We thank the Reviewer for this comment. Over the years, we have thoroughly discussed this crucial point of the work with the multicenter and international group of urologists, oncologists, and pathologists who co-authored this work.

Here our conclusions:

i. qPCR data show remarkable differences between endogenous and exogenous TRPM8 RNA levels in prostate cells. This is particularly true for RWPE-1 where the fold induction between the exogenous and endogenous TRPM8 transcripts are dictated by the very low expression of the endogenous RNA as demonstrated by the CT reported below. This sharp dichotomy between the levels of TRPM8 transcript and protein is not peculiar of RWPE-1 cells. We have recently demonstrated that in human and also genetically engineered mouse models of metastatic and castration resistant prostate cancer high levels of the TRPM8 channel are frequently associated with very low levels of the RNA (Genovesi et al., 2022). A manuscript in preparation demonstrates the same condition in Lung, Breast and Colorectal cancers where the amount of TRPM8 RNA is minimal (sometimes below the detection range by RNAseq and qPCR) while TRPM8 channel is consistently expressed (sometimes at very high levels) and functional. A speculative hypothesis to reconcile these findings stems on the concept that while the increase amount and activity of the channel could support tumorigenesis, the rise of TRPM8 RNA stimulate TLR3 inflammation and the recruitment of NK cells, which obviously should be avoided by the tumor cells. In support of that, higher expression of TRPM8 RNA is a predictor of good prognosis in PCa (Supplementary Fig. S1f in Alaimo et al., 2020). That said, the amount of exogenous TRPM8 transcripts (M8 and MM) in prostate cells is far from unnatural. In human tumours (TCGA) the levels of TRPM8 are generally comparable (0.5) to the ones of GAPDH, a commonly used housekeeping transcript. Consistently, in prostate cells (RWPE-1 and LNCaP) the TRPM8/GAPDH ratios in the overexpression models (M8 and MM) are more similar to the tumour rather than WT cells.

RWPE-1

TRPM8 WT 34/35ct/GAPDH 19/20ct

TRPM8 M8 21/22ct/GAPDH 19/20ct

TRPM8 MM 16/17ct/GAPDH 19/20ct

LNCaP

TRPM8 WT 26/27ct/GAPDH 19/20ct

TRPM8 M8 20/21ct/GAPDH 19/20ct

TRPM8 MM 16/17ct/GAPDH 19/20ct

Importantly, results of digital droplet PCR (ddPCR) analysis (new Figure 2C) demonstrates that exogenous TRPM8 MM transcript in LNCaP MM cells is approximately 50 times more abundant than endogenous TRPM8 RNA in LNCaP wild type. In addition, two newly generated stable LNCaP cell lines (MM*) with a fold-change by qPCR of less than 50 between exogenous (MM) and endogenous TRPM8 transcript, retain nuclear accumulation of NF- κ B and induction of KLK2 gene expression (new Figure EV2E and F).

Finally, the fold change between MM and endogenous TRPM8 transcripts in LNCaP xenografts is about 16, and the CT are comparable to the transcript of the house keeping gene ACTB.

LNCaP Xenografts

TRPM8 WT 31 Ct/ACTB 23-24 Ct

TRPM8 MM 28 Ct/ACTB 24-25 Ct

- ii. Knock-down and knock-out experiments presented in the manuscript prove the ability of the endogenous TRPM8 transcript to modulate TLR3/NF- κ B activation in prostate cells. Specifically:
- Fig 1A and D: CAS vs WT
 - Fig 1B-C: siTRPM8 vs siCtr
 - Fig 2D,E,F,G; EV2D and F: CAS vs WT
 - Fig 3D,E,J; EV3B: CAS vs WT

This set of data defines the cellular models used in this work as a valuable tool to decipher the molecular circuits connecting TRPM8 transcript expression to sterile inflammation and prostate pathology. Notwithstanding, we would like to emphasize that one of the goals of our work is to pave the way for a potential clinical strategy to promote NK tumour suppressor activity in PCa (and maybe in other types of solid malignancies) via the administration of TRPM8 derived RNA oligos that we are in the process of identifying and patenting. In this regard, overexpression of TRPM8 RNA is functional for the goal, regardless of the physiological level of the transcript in healthy and malignant tissues.

2) The manuscript in general would benefit from a more clear hypothesis and some re-organization to provide a more streamlined connection between TRPM8-Tlr3-Nf-Kb.

The implications of increased PSA upon TRPM8 RNA increase in exosomes should also be stated more clearly.

Both TAK and TLR3 inhibitors could have effects on the NFkB pathway independent of TRMP8 or in response to other components of the exosomes. A direct connection of TRPM8 RNA with TLR3 does not appear to have been established.

Supernatant experiments could be performed with TLR3i to strengthen this point.

What other components of the NF-Kb pathway are activated?

In addition, NF-KB is part of numerous pathways, are other inflammatory pathways activated?

We totally agree with our Reviewer that a streamlined connection between TRPM8/TLR3/NF-kB is missing. This work started in 2018 with the serendipitous finding of PSA expression in RWPE-1 cells stably expressing exogenous TRPM8 cDNA. Since then, we have spent an enormous amount of time and effort trying to identify a molecular connection that exclusively links TRPM8 to PSA expression through androgen receptor activity. After an endless list of failures, we decided to browse the literature in search of other transcription factors capable of controlling PSA expression independently of AR and found Sawyers paper about NF-kB and PSA (Chen & Sawyers, 2002). Then we analysed NF-kB and identified the nuclear shuttling of NF-kB in cells overexpressing TRPM8. Once excluded the channel activity as a driver of NF-kB activation, we moved our attention to the RNA and RNA sensors connected to NF-kB. As our Reviewer can easily imagine, this has been a very long and puzzling story, which nevertheless was extremely exciting as only serendipity can do.

We state more clearly in the Discussion the possible clinical implication of TRPM8 RNA dependent modulation of PSA expression.

As suggested by the Reviewer, the revised manuscript includes the new Fig EV4E showing the supernatant experiment with the administration or not of TLR3i and a broader analysis of pro-inflammatory effectors. The result strengthens the functional link between TRPM8 RNA secretion in EVs (new Fig 3E) and TLR3-dependent activation of pro-inflammatory pathways, included IRF3/IFN β /STAT1.

Furthermore, to investigate the possible role of other RNA sensors -besides TLR3- in the activation of NF-kB, we tested TLR7/8, RIG1 and PKR inhibitors. Results are showed in Fig 3G and Fig EV4F and demonstrate the exquisite role of TLR3 in TRPM8 RNA sensing and activation of pro-inflammatory pathway. These data confirm the fact that TLR3 inhibition is sufficient to turn off the inflammatory signals.

3) A lack of effect of MM translation-defective mutant on proliferation and cell death is claimed, however, only cell cycle analyses by FACS are shown. Images of cell cultures for morphology, growth curves and some cell death assays should be performed to support this conclusion. The authors performed similar assays in a previous publication where they characterized the effects of the M8 overexpression, but the MM mutant has not been previously characterized.

In line with the reviewer's suggestion, the revised manuscript includes the Appendix Fig S1 showing a wide panel of assays to carefully characterize cell features associated with the inflammatory condition. As previously described for overexpression of the TRPM8 coding RNA (M8) in prostate cells (Alaimo et al., 2020), expression of the non-coding transcript does not impose specific phenotypes on the prostate cells used for this work.

4) Major conclusions are drawn based on the MM mutant. Does this RNA have the same conformation as the WT RNA in terms of secondary structure? Could the mutant be more immunogenic? If this is the case, this should be addressed.

The predicted secondary structure of MM TRPM8 RNA is provided. As shown by the images, mutation of the two nucleotides at the 5-prime end of the transcript does not affect the secondary structure.

Wild Type human TRPM8 CDS

MM mutant human TRPM8 CDS

5) Trichrome staining in Fig. 4D should be supported by quantification, same for SMA and vimentin in Fig. 4E, especially since vimentin levels appear increased in the WB of MM xenografts in Fig. 4C.

We agree with our Reviewer that quantification of type I Collagen, Vimentin and SMA in the xenograft sections can contribute to improve the analysis and strength the result. Quantification of Collagen staining on histologic section of xenografts is described in Fig EV5D while quantification of Vimentin and SMA is reported in Fig EV5 E, the latter confirming the point raised by the Reviewer about the increased amount of Vimentin expression in fibroblasts infiltrating the MM xenografts.

6) While the LNCap cell lines might generate EVs containing TRPM8 RNA, the TRPM8 RNA in blood exosomal vesicles might not be derived from cancer cells. Can the source/origin of blood exosomal TRPM8 be addressed? Is it increased in tumour cells from the same patients as well?

We thank the Reviewer for raising an important observation, particularly relevant in the field of liquid biopsies. Defining the tissue of origin of EVs is not trivial since, as the Reviewer points out, EVs and their RNA cargo may potentially derive from any cell of the organism. Despite the possibility of estimating the relative contribution of different tissues to the total signal in the circulation via supervised deconvolution (Stephen R. Quake, 2018, *Nature Biotechnology*), determining the origin of a single transcript is still an open question in the field and, to our knowledge can't be addressed directly leveraging liquid biopsy data alone.

While we don't have access to matched tissue biopsies or circulating tumor cells (CTCs) from the same patients as our liquid biopsy collection is part of a biofluid biobank, there is supporting evidence that circulating TRPM8 derives from patients' cancer cells:

- TRPM8 is a prostate-specific transcript. TRPM8 is expressed almost exclusively from prostatic tissue – the liver is the only other organ that potentially expresses TRPM8, albeit at significantly lower levels (<https://gtexportal.org/home/> , Uhlén M et al., *Tissue-based map of the human proteome*, 2015, *Science*, DOI: 10.1126/science.1260419).

- TRPM8 is highly expressed in prostate cancer, also with respect to other cancer types (TCGA data, <https://www.proteinatlas.org/ENSG00000144481-TRPM8/pathology>).

- TRPM8 is absent in the circulation of male, age-matched healthy donors (HD), indicating that, as expected, benign prostate (and liver) do not contribute to TRPM8 signal in the circulation.

- TRPM8 transcript is detected in prostate cancer patients' circulation, whether or not they underwent prostatectomy (no significant difference between the two groups, prostatectomy Yes (n=15), prostatectomy No (n=12), $p=0.58$ Wilcoxon's test), thus indicating that the circulating signal derives from prostate cancer metastases (observations and data will be included in the revised manuscript).

Minor critiques:

7) The abstract text would benefit from some rewording to increase clarity especially in regard to the connection of the sterile inflammation to cancer.

We modified the Abstract to better describe the relevance of sterile inflammation in cancer.

8) Expanded Figure 1D (Fig. S1D): is PSA WB from the same probing? If this is from a different WB, this should be stated.

We stated in the figure legend that GAPDH was analysed after stripping of the primary and secondary antibodies used for PSA.

9) Fig. 2E: since RWPE1 cells do not express detectable Nkx3-1 levels, another unchanged marker should be shown.

NKX3.1 was chosen because not expressed in RWPE-1 exactly as PSA, which is induced by TRPM8 unlike NKX3.1. The revised Fig 2E now includes KLK2.

10) Expanded Figure 2: PC3 cells are shown, but not mentioned in the main text.

We apologize for this oversight. We amended the text.

11) Fig. 3D, 3E: WT supernatant is mentioned in the text, however only supernatant from CAS, M8, and MM are shown. Could this be clarified? Expanded Figure 3A shows the LNCap WT supernatant results, but not the WT RWPE1 supernatant.

In the revised text, we specified that WT refers only to LNCaP supernatant. Wild type RWPE-1 cells express very low levels of TRPM8 transcript according to qPCR data and they do not promote NF- κ B activation and KLK3 expression.

RWPE-1

TRPM8 WT 34/35 Ct-GAPDH 19/20 Ct

LNCaP

TRPM8 WT 26/27ct/GAPDH 19/20ct

12) Could the Nf-Kb p65 nuclear shuttling be supported by imaging? Are all cells responding in the same way?

As shown by Fig 2I, immunolocalization of NF- κ B in wild type and MM RWPE-1 cells shows nuclear NF- κ B in the vast majority of MM cells compared to the primarily cytosolic distribution of NF- κ B in WT LNCaP.

13) Fig 4C: a loading control appears to not have been included.

We included β -Actin as loading control of the WB showed in Fig 4C.

14) Is there a hypothesis behind including NAFs and CAFs? This should be made more clear in the text. What is the explanation for an effect of MM supernatant on CAFs, but not on NAFs in Fig. 4H? The changes in CAFs are more consistent, but some (such as CXCL10) appear modest in CAFs compared to NAFs. Is there a conclusion/suggestion to be made of these results?

One of the crucial steps of the tumorigenic process is the transformation of the microenvironment. Normal prostate tissue is largely composed by fibroblasts and stromal reaction is a hallmark of malignant transformation. Secreted signals by tumour cells are the main suspects of stromal changes but the molecular determinants and the signalling pathways are still poorly understood. The rationale to test both NAFs and CAFs in Figure 4H was to investigate the possible implication of TRPM8/TLR3 signalling in NAF to CAF transition, and, on the other hand, in CAF biology.

We agree with our Reviewer that the induction of CXCL10 in NAFs conditioned with the supernatant from LNCaP MM cells is interesting. So far, there is no explanation for this result, only speculative suggestions pointing to the synergism of NF- κ B and STAT1 activities in fibroblasts for the induction of CXCL10 expression. IFN- β expression increases in LNCaP and RWPE-1 MM cells likely inducing STAT1 activation in fibroblasts conditioned with the supernatants from MM prostate cells. It can be speculated that the IFN- β /STAT1 pathway may be functional in NAFs while partially defective in CAFs.

15) How are NF- κ B and TLR3 affected in MM or ME xenografts?

We tested 2 different commercial antibodies and two different antigen retrieval protocols on sections of paraffin embedded pellets of LNCaP and RWPE-1 CAS and MM cell lines used as negative (CAS, cytosolic staining) and positive (MM, nuclear staining) controls. However, neither of the two antibodies [NF- κ B p65 (D14E12) #8242 from Cell Signaling (perfectly working for WB and IF), and the Santa Cruz Biotechnology p-NF- κ B p65 (A-8): sc-166748] showed a staining in MM (nuclear) and CAS (cytosol) control samples.

Regarding TLR3, Western blot analysis is now included in the revised version of Fig 4C and shows similar levels of protein expression in wild-type and MM LNCaP xenografts. Immunolocalization analyses was performed on FFPE sections with three independent antibodies (mouse monoclonal NBP2-24875; rabbit polyclonal PA5-20183, and the rabbit polyclonal NB100-56571). NBP2-24875 stained the nucleus of both wild-type and MM LNCaP xenografts, PA5-20183 marked the cytosol and nucleus in both samples, while the NB100-56571 mildly stained the cytosol, as showed below. Although the staining with secondary Ab alone was completely negative, without a better control -e.g. FFPE section of TLR3-null LNCaP xenografts- we would prefer not to include these data in the revised manuscript.

16) While the tumours appear histologically different, there seems to be no change in tumour burden. Could some treatments be included in this study to strengthen the clinical implications?

Xenografts with LNCaP cells took almost 4 months to be palpable. Although the point raised by our reviewer is of undoubted interest, this experiment is not feasible in the time frame of a revision. However, a main project in the lab has been funded by the Italian Association for Cancer Research (AIRC) this year to investigate the relevance of TRPM8 pharmacologic targeting in vivo. A key aspect of the project is the generation of orthotopically transplanted mouse models with human (nude mice) and mouse (syngeneic) PCa cell lines expressing the endogenous gene or the exogenous cDNA encoding for the TRPM8 channel (M8). Treatment with potent TRPM8 agonist will be combined with standard hormone and chemo (taxanes) therapies to investigate possible synergisms. Since M8 cell lines trigger the NF- κ B response, we will include in the experimental plan the assessment of the contribution of TRPM8 RNA-induced sterile inflammation to anticancer treatments efficacy.

17) TRPM8 has been shown to exert an inhibitory effect on cell migration. Does TRPM8 RNA play a role in cell migration?

According to the data collected in the new Appendix Fig S1, overexpression of TRPM8 transcript does not promote migratory and invasive properties in LNCaP cells. Similar results were obtained in LNCaP and RWPE-1 cells with the overexpression of the coding transcript (M8) or the knock-out of the gene by Crispr/Cas9. We cannot exclude to obtain different results with the pharmacologic modulation of channel activity.

18) In the de Souza reference included in the text, although TRPM8 RNA is present in blood, there is no significant difference by qPCR between TRPM8 RNA in prostate cancer patients blood compared to normal blood. How are the blood EVs results from this manuscript related to the de Souza results? Are blood EVs enriched for TRPM8 RNA in prostate cancer patients?

We thank our Reviewer for pointing out this important aspect that most likely intersects the still poorly understood biology of extracellular vesicles. de Souza and colleagues described the TRPM8 RNA as a freely circulating transcript in the blood of both healthy donors and PCa patients. No difference in the amount was detected at that time between the two groups (PLOS ONE 2017). In a follow up article published in 2020, the same group define higher amounts of circulating free TRPM8 RNA in the blood of PCa patients as a predictor of high-risk tumours (Carcinogenesis 2020, our reference). Our data on human EVs was derived from a large multicentric study led by Prof. Francesca Demichelis's group, an internationally renowned computational scientist in the field of PCa. According to these data, TRPM8 RNA is significantly enriched in EVs collected from the blood of PCa patients compared to those extracted from the blood of healthy donors. Of note, two different approaches demonstrate the presence of TRPM8 RNA circulating in the blood of humans. Moreover, interesting discussions in a recent workshop (QuantitatEVs: Multiscale analyses, from bulk to single vesicle, of the International Society of Extracellular Vesicles, Trento 2023) point to the enrichment of transcripts encoding for plasma membrane-associated and transmembrane

proteins into the EVs. This could help reconcile the two aspects related to the circulation of TRPM8 RNA in the blood.

Minor suggestions:

The authors should check for few typos, there are also few paragraphs missing some words or having some extra words. For instance, the last sentences on pg. 7 need some re-writing.

We thank our Reviewer for the advice.

Referee #2:

In the manuscript entitled, "Androgen-independent PSA spikes unmask a novel mechanism of sterile inflammation with critical roles in prostate tumorigenesis," the authors outlined a novel inflammatory mechanism in the prostate epithelium caused by the interaction of TRMP8 RNA with TLR3. They show that TRMP8 regulates KLK3 (PSA) expression in an AR-independent mechanism in both immortalized and prostate cancer cell lines. They demonstrate that enforced expression of TRMP8 RNA regulates NF- κ B activation and PSA induction without interfering with cell proliferation and survival. Furthermore, they found that TRPM8 RNA contributes to NF- κ B activation through endosomal TLR3 signalling through translational-independent mechanisms within extracellular vesicle (EVs). As a result of TRPM8-induced inflammation in PCa xenografts, NK cells infiltrated more readily, and cancer-associated fibroblasts (CAFs) decreased collagen type I deposition, likely contributing to anti-tumour immunity.

While the mechanism is quite interesting, the heavy dependence on an overexpression system raises the important question of physiologic relevance. We point out specific areas in this manuscript that the authors can consider to improve this manuscript:

We thank our Reviewer to point it out the interest in the novel mechanism of sterile inflammation unravelled by our work. Treasuring the criticisms raised by the Reviewer, we consolidated our findings and strengthened our conclusions.

Over the years, we have thoroughly discussed this crucial point of the work with the multicenter and international group of urologists, oncologists, and pathologists who co-authored this work.

Here our conclusions:

iii. qPCR data show remarkable differences between endogenous and exogenous TRPM8 RNA levels in prostate cells. This is particularly true for RWPE-1 where the fold induction between the exogenous and endogenous TRPM8 transcripts are dictated by the very low expression of the endogenous RNA as demonstrated by the CT reported below. This sharp dichotomy

between the levels of TRPM8 transcript and protein is not peculiar of RWPE-1 cells. We have recently demonstrated that in human and also genetically engineered mouse models of metastatic and castration resistant prostate cancer high levels of the TRPM8 channel are frequently associated with very low levels of the RNA (Genovesi et al., 2022). A manuscript in preparation demonstrates the same condition in Lung, Breast and Colorectal cancers where the amount of TRPM8 RNA is minimal (sometimes below the detection range by RNAseq and qPCR) while TRPM8 channel is consistently expressed (sometimes at very high levels) and functional. A speculative hypothesis to reconcile these findings stems on the concept that while the increase amount and activity of the channel could support tumorigenesis, the rise of TRPM8 RNA stimulate TLR3 inflammation and the recruitment of NK cells, which obviously should be avoided by the tumor cells. In support of that, higher expression of TRPM8 RNA is a predictor of good prognosis in PCa (Supplementary Fig. S1f in Alaimo et al., 2020). That said, the amount of exogenous TRPM8 transcripts (M8 and MM) in prostate cells is far from unnatural. In human tumours (TCGA) the levels of TRPM8 are generally comparable (0.5) to the ones of GAPDH, a commonly used housekeeping transcript. Consistently, in prostate cells (RWPE-1 and LNCaP) the TRPM8/GAPDH ratios in the overexpression models (M8 and MM) are more similar to the tumour rather than WT cells.

RWPE-1

TRPM8 WT 34/35ct/GAPDH 19/20ct

TRPM8 M8 21/22ct/GAPDH 19/20ct

TRPM8 MM 16/17ct/GAPDH 19/20ct

LNCaP

TRPM8 WT 26/27ct/GAPDH 19/20ct

TRPM8 M8 20/21ct/GAPDH 19/20ct

TRPM8 MM 16/17ct/GAPDH 19/20ct

Importantly, results of digital droplet PCR (ddPCR) analysis (new Figure 2C) demonstrates that exogenous TRPM8 MM transcript in LNCaP MM cells is approximately 50 times more abundant than endogenous TRPM8 RNA in LNCaP wild type. In addition, two newly generated stable LNCaP cell lines (MM) with a fold-change by qPCR of less than 50 between exogenous (MM) and endogenous TRPM8 transcript, retain nuclear accumulation of NF- κ B and induction of KLK2 gene expression (new Figure EV2E and F).*

Finally, the fold change between MM and endogenous TRPM8 transcripts in LNCaP xenografts is about 16, and the CT are comparable to the housekeeping transcript of the house keeping gene ACTB.

LNCaP Xenografts

TRPM8 WT 31 Ct/ACTB 23-24 Ct

TRPM8 MM 28 Ct/ACTB 24-25 Ct

- iv. Knock-down and knock-out experiments presented in the manuscript prove the ability of the endogenous TRPM8 transcript to modulate TLR3/NF- κ B activation in prostate cells. Specifically:
Fig 1A and D: CAS vs WT
Fig 1B-C: siTRPM8 vs siCtr
Fig 2D,E,F,G; EV2D and F: CAS vs WT
Fig 3D,E,J; EV3B: CAS vs WT

This set of data defines the cellular models used in this work as a valuable tool to decipher the molecular circuits connecting TRPM8 transcript expression to sterile inflammation and prostate pathology. Notwithstanding, we would like to emphasize that one of the goals of our work is to pave the way for a potential clinical strategy to promote NK tumour suppressor activity in PCa (and maybe in other types of solid malignancies) via the administration of TRPM8 derived RNA oligos that we are in the process of identifying and patenting. In this regard, overexpression of TRPM8 RNA is functional for the goal, regardless of the physiological level of the transcript in healthy and malignant tissues.

Major Comments:

1) The authors claim that "Unexpectedly, the results were recapitulated even under complete androgen blockade in Fig 1D." However, KLK3 and other AR transcriptional targets are not affected by LNCaP WT Enzalutamide treatment. This is a very peculiar findings and goes against published literature.

We apologize if the experimental setup was not sufficiently clear. LNCaP cells were grown in charcoal serum, a serum deprived of steroid hormones, as outlined in the figure legend. Enzalutamide was used to ensure complete absence of residual AR activity. The lack of reduction in the expression of androgen-responsive genes in the presence of Enzalutamide was expected due to charcoal serum but further proved the AR-independency of KLK3 expression in prostate cells overexpressing TRPM8.

We labelled revised Fig 1D with "Steroid-stripped charcoal serum".

2) In Fig 1H-I, the authors conclude that "the PSA amount in both LNCaP and RWPE-1 cells was not influenced by gating TRPM8 channel with its potent agonist WS-12 or antagonist AMBT (Fig 1H and I), suggesting that channel activity is not involved in the observed effects associated with TRPM8 overexpression." Since TRPM8 is a Ca²⁺ uptake channel, the authors should quantify the Ca²⁺ efflux response to conclude no involvement of TRPM8 channel activity in PSA regulation. This can be efficiently done using fluorescent calcium indicators like Fura-2 AM and Fluo-4.

We agree with our Reviewer that direct evidence of TRPM8 activity can strengthen the conclusion that KLK3 expression is not dependent by the function of the channel. Calcium currents in RWPE-1 and LNCaP expressing endogenous, exogenous, or not expressing (CAS) TRPM8 channel have been deeply characterized in a previous paper published by our group in 2020 (Alaimo et al., Cell Death and Disease). Ca²⁺ currents have been measured through FURA-2 and patch clamp analyses. RWPE-1 cells expressing exogenous levels of TRPM8 showed robust calcium intake in response to channel agonists. Ca²⁺ currents were not measurable in LNCaP and RWPE-1 cells expressing endogenous levels of TRPM8 nor in LNCaP M8 cells expressing exogenous TRPM8. Nevertheless, increased amount of calcium in these cells was evident through biochemical studies showing CaMKII activation and cellular phenotyping demonstrating calcium cytotoxicity. For a more accurate assessment of the results, please refer to Alaimo et al., CDDis 2020: "In prototypes of primary and metastatic PCa recapitulating high TRPM8 expression, Ca²⁺ cytotoxicity induced by potent TRPM8 agonists combined with a sublethal dose of X-rays generates an overwhelming cellular stress that overcomes the anti-apoptotic barriers established in cancer cells by the impairment of PTEN/PI3K/AKT axis. Still, the metastatic progression of the disease is frequently associated with complete loss of PTEN in prostate tumour cells⁵⁰, which has been demonstrated to counteract Ca²⁺-dependent cell death by favouring the proteasome degradation of IP3R³⁹. In the PTEN-null hormone naïve lymph node metastatic prostate cell line LNCaP⁴³, Vanden Abeele and colleagues describe the IP3R1 as the IP3 receptor preferentially expressed, followed by almost three times lower levels of IP3R3 and very low, if any, expression of IP3R2⁵¹. Dysfunctional IP3Rs make TRPM8 agonists unable to induce a rapid and massive Ca²⁺ store depletion in LNCaP, which is the cornerstone of Ca²⁺-induced cytotoxicity. However, different by icilin that induces extensive desensitization of the channel, TRPM8 activation through either menthol or WS-12 is followed by moderate channel adaptation^{29,52,53}. This important aspect regarding TRPM8 pharmacology allows us to extend the analysis to longer time points and demonstrate that 48 h of WS-12 treatment doubles the percentage of apoptotic cells in LNCaP. This result together with the phosphorylation of CaMKII on threonine 286 suggests that prolonged pharmacological activation of TRPM8 can trigger cytotoxicity, possibly via a smoothly graded increase of [Ca²⁺]_i in LNCaP cells, which is likely contributed by continual quantal releases of Ca²⁺ from the ER⁵⁴⁻⁵⁸. Overall, these data demonstrate that extracellular and/or intracellular stimuli leading to minimal increases of [Ca²⁺]_i can still determine an harmful cellular stress if prolonged over time. Indeed, 48 h of WS-12 treatment combined with either docetaxel or enzalutamide rises the percentage of cell death in LNCaP cells overexpressing the channel from roughly 20% with single treatments to almost 60%."

Finally, in support of the fact that channel activity appears to be irrelevant to PSA expression, the translationally defective MM TRPM8 transcript produces similar effects on PSA levels without encoding the channel.

3) In Fig 3D and E, the authors show that supernatants of WT, M8, and MM containing TRPM8 RNA activates NFkB signalling. However, they do not show cytosolic and nuclear NF-kB p65 levels in LNCaP and RWPE-1 cells without supernatant.

Revised Fig 1G and H show the nuclear/cytosolic distribution of NF-kB in LNCaP and RWPE-1 CAS, WT, MM, and M8 lines without the conditioning of supernatants.

4) Also why did the authors use supernatant in Fig 3D and E instead of purified EVs? This is a major confounding factor of this experiment because supernatant is rich with many other things besides EVs.

We agree with our Reviewer on the necessity to separate EVs from the supernatants and test in parallel the two fractions to prove the relevance of EV-entrapped TRPM8 RNA in TLR3 signalling and NF-KB activation. The revised Fig 3E described the results obtained.

5) In Fig EV3C and D, the authors show in silico prediction of TRPM8 RNA structure with stable stems and loops. However, this does not confirm that these structures of TRPM8 RNA binds to TLR3. Since this is one of the key conclusions of the proposed model, this needs to be proven experimentally. The authors could pull down the TLR3/ligand complex and RT for TRMP8 RNA.

This was a very challenging experiment, as validated protocols and antibodies for TLR3 were not described in the literature. Therefore, we are very pleased to present in Fig 3H-J, EV3A and Appendix Fig S1 the new RIP and fCLIP data demonstrating the binding of different fragments of the endogenous and exogenous TRPM8 transcript to TLR3.

6) In Fig 4F authors show that the CAFs reduces COL1A1 protein when conditioned with MM supernatant. Is this effect only due to TRPM8 MM RNA? Can this be an indirect effect because of other secretory differences besides having more TRMP8 RNA?

This is another very important point. We cannot rule out that CAFs response to MM supernatants might be dependent on other secreted factors and not on TRPM8 RNA. However, as shown by the new Fig EV5I and J, administration of TLR3 inhibitor to CAFs conditioned with LNCaP cells supernatants fully rescues the expression of PH3, COLGALT1 and COLGALT2 genes in CAFs and the amounts of COL1A1 protein. These new data support the thesis of TRPM8 RNA/TLR3 signalling in the alteration of collagen synthesis and deposition by CAFs.

7) The authors confirmed that LNCaP MM xenografts show a higher amount of PSA in the tissue however the circulating total (free + complexed) PSA was the opposite. What does this mean and how does it fit the proposed model?

The reverse correlation between intratumor and circulating PSA was one of the aspects of the work that sparked the most interest among our clinical collaborators. Circulating PSA is critical to track tumour response to therapy. Robust reduction of the amount of circulating PSA signifies a good response of the tumour to therapy and favourable prognosis. On the other hand, a slight reduction

in, or the rise of, circulating PSA is generally interpreted by urologists and oncologists as a poor response to therapy (or a biochemical recurrence) and a strong indication to move patients to more aggressive treatments. TRPM8 RNA influence on PSA protein (not the transcript) can result highly confounding in this scenario. If on the one hand by increasing PSA protein levels independently on AR activity this might generate false negative response to hormone therapy, on the other hand by entrapping PSA into the tumour mass TRPM8 RNA can alter both the diagnostic -poor- and prognostic -high- value of circulating PSA since this measure is no more representative of AR activity in cancer cells. Thanks to the proof of concept generated by this work, the overall conclusion of our team of urologists and pathologists is that circulating PSA quantification should be coupled with quantitative immunolocalization analyses on prostate tumour samples to avoid misleading interpretations of tumour response to therapy indicated by circulating PSA.

Minor comments

1. On page 5 authors say, "Once stably overexpressed in both TRPM8 knocked-out (CAS) LNCaP and RWPE-1 cell lines, the TRPM8 mutant transcript (MM) was shown to be stable (Fig 2B) but unable to encode for the protein (Fig EV2A-C)." This sentence needs restructuring.

We thank the Reviewer for this suggestion. Here is the new sentence, 'TRPM8 mutant transcript (MM) was stable but unable to encode for the protein in both LNCaP and RWPE-1 TRPM8 knock-out (CAS) cell lines and in PC3 cells (Fig 2B, Fig EV2A-C).'

2. The schematic of potential mechanism of TRPM8 RNA/TLR3 molecular circuit in Fig 3C should be at the end of Fig 3. So that the authors have shown the TLR3 NFkB activation related experiments before presenting the model.

The vignette in Fig 3C represents the working model we hypothesized to explain the link between TRPM8 RNA secretion in EVs and NF-kB activation. Although we agree that it might be useful to summarize the results in Figure 3, we would prefer to leave the vignette at the beginning of Figure 3 to help readers understand the rationale behind the experimental workflow.

3. In general, these figures and their legends need additional labelling and experimental details: Fig EV2D, Fig EV2E, Fig 3A, Fig 3B, Fig 4B, Fig 4D.

We improved the labelling and legend of the Figures indicated by the Reviewer.

Dear Dr Lunardi,

Thank you for submitting your revised manuscript (EMBOJ-2023-114493R) to The EMBO Journal. My sincere apologies for the unusual delay in assessing your amended manuscript, which was due to protracted referee input as well as detailed discussion in the editorial team. Your complemented study was sent back to the referees for their re-evaluation, and we have received comments from both of them, which I enclose below.

As you will see, referee #1 stated that the work has been substantially improved by the revisions and s/he is now in favour of publication. Referee #2 remains overall more critical. However, we have discussed the input and concluded that in light of the support of referee #1 and the additional data provided in support of your claims, we can give you the opportunity to further adjust the manuscript in another round of revision. We are then prepared to swiftly proceed towards acceptance and publication of the work.

Please consider the remaining points by the referees carefully and add complementary data to address the critique, or adjust the text where appropriate, toning down the claims made.

Also, we now need you to take care of a number of minor issues related to formatting and data presentation as detailed below, which should be addressed at re-submission.

Please contact me at any time if you have additional questions related to below points.

As you might have noted on our web page, every paper at the EMBO Journal now includes a 'Synopsis', displayed on the html and freely accessible to all readers. The synopsis includes a 'model' figure as well as 2-5 one-short-sentence bullet points that summarize the article. I would appreciate if you could provide this figure and the bullet points.

Thank you for giving us the chance to consider your manuscript for The EMBO Journal. I look forward to your final revision.

Again, please contact me at any time if you need any help or have further questions.

Kind regards,

Daniel Klimmeck

>> Please limit the number of keywords for your study to maximally five.

>> Abstract: please limit to maximally 175 words.

>> Rename the current 'Conflict of Interest' section to 'Disclosure and Competing Interests Statement'.

>> Author Contributions: Please remove the author contributions information from the manuscript text. Note that CRediT has replaced the traditional author contributions section as of now because it offers a systematic machine-readable author contributions format that allows for more effective research assessment. and use the free text boxes beneath each contributing author's name to add specific details on the author's contribution.

More information is available in our guide to authors.

>> Appendix: please provide page numbers in the ToC on its first page: remove yellow highlights.

>> Callouts: revisit the callout for Fig.S1D in the main text (referring to Fig EV1D?)

>> Author Checklist: specify cell line authentication status of the cell lines used.

>> Tables EV1 and 2 are uploaded as a reagents list. Please upload as two separate files labelled Table EV1 and Table EV2.

>> Data availability section: please move the published GEO dataset annotation here. Add a sentence stating: 'No additional large-scale data amenable to public repository deposition were generated in this study.' .

>> Source data: the uploaded source data does not correspond to the current uploaded figures. They need to be updated to the current figure set. My colleague Hannah Sonntag (CC'ed in) will contact you shortly in a separate message on the matter.

>> Consider additional changes and comments from our production team as indicated below:

- Figure legends:

1. Please note that a separate 'Data Information' section is required in the legends of all the figures.
2. Please indicate the statistical test used for data analysis in the legends of figures 1a, b, d, e; 2d, f; 3b; 4b, g, m; EV1c, e; EV2d, g, h; EV5b-e, i, k, l
3. Please define the annotated p values ***/**/* in the legend of figure EV5b-e, i, k, l as appropriate.
4. Please note that the box plots need to be defined in terms of minima, maxima, centre, bounds of box and whiskers, and percentile in the legends of figures 3b; 4i-l; EV5b, c, d, e
5. Please note that information related to n is missing in the legends of figures EV3a; EV5d, e, h, i, k, l
6. Please note that the error bars are not defined in the legends of figures EV3a; EV5h, i, k, l

- Data citation:

Please note that the data citation callouts in text do have any matching entries in the Reference list.

Referee #1:

The findings presented in the revised manuscript contribute valuable insights to the field. The authors included new experiments/figures and text revisions that addressed my comments appropriately. Minor suggestions: The authors should carefully go over the figure legends (including the supplemental fig. legends) to include all the relevant statistical details and over the text for the few remaining typos.

Referee #2:

1) The authors do not provide enough evidence that their overexpression of TRPM8 mRNA is physiologic. What they need to show is how much TRPM8 mRNA there is between normal and cancer cells. Or perhaps less aggressive versus more aggressive cancer cells. The siRNA experiment in 1B-C does not prove that it is the transcript (and not the protein) that drives the phenotype.

2) Their response to R2 Q3 - Unclear what they are referring to - wrong references - New 3E? Where is the data for RWPE-1 (+/- EV)?

3) Their response to R2 Q6 - This experiment does not answer the question. The question is that conditioned media might contain a factor besides TRPM8 containing EVs that drive this process. One way to address this question would be to use EV deplete conditioned media.

4) Their response to R2 Q6 - This response is incorrect and not consistent with the current use of PSA in the clinic. PSA responses (in the blood) to androgen deprivation therapy are nearly universal. Furthermore, the results remain puzzling.

**UNIVERSITÀ
DI TRENTO**

**Department of
Cellular, Computational and Integrative Biology - CIBIO**

Manuscript EMBOJ-2023-114493R

Dear Dr Klimmeck,

The following are all the formatting changes included in the revised version of the manuscript and the point by point rebuttal to the Reviewers' comments:

Please limit the number of keywords for your study to maximally five

Done

Abstract: please limit to maximally 175 words

Done

Rename the current 'Conflict of Interest' section to 'Disclosure and Competing Interests Statement'

Done

Author Contributions: Please remove the author contributions information from the manuscript text. Note that CRediT has replaced the traditional author contributions section as of now because it offers a systematic machine-readable author contributions format that allows for more effective research assessment. and use the free text boxes beneath each contributing author's name to add specific details on the author's contribution

Done

Appendix: please provide page numbers in the ToC on its first page: remove yellow highlights

Done

Callouts: revisit the callout for Fig.S1D in the main text (referring to Fig EV1D?)

not found

Author Checklist: specify cell line authentication status of the cell lines used

Done

Tables EV1 and 2 are uploaded as a reagents list. Please upload as two separate files labelled Table EV1 and Table EV2

Done

Data availability section: please move the published GEO dataset annotation here. Add a sentence stating: 'No additional large-scale data amenable to public repository deposition were generated in this study.'

Done

Source data: the uploaded source data does not correspond to the current uploaded figures. They need to be updated to the current figure set

Done

My colleague Hannah Sonntag (CC'ed in) will contact you shortly in a separate message on the matter.

Point by point rebuttal

We are honestly grateful to our Reviewers for their insightful comments and suggestions that substantially improved the quality and robustness of our findings and strengthened our conclusions. We hope that the manuscript will now be considered suitable for publication in EMBO Journal.

Referee #1:

The findings presented in the revised manuscript contribute valuable insights to the field. The authors included new experiments/figures and text revisions that addressed my comments appropriately. Minor suggestions: The authors should carefully go over the figure legends (including the supplemental fig. legends) to include all the relevant statistical details and over the text for the few remaining typos.

We are very grateful to our Reviewer, we will include relevant statistical details in the figure legends and correct the remaining typos in the text.

UNIVERSITÀ
DI TRENTO

Department of
Cellular, Computational and Integrative Biology - CIBIO

Referee #2:

1) The authors do not provide enough evidence that their overexpression of TRPM8 mRNA is physiologic. What they need to show is how much TRPM8 mRNA there is between normal and cancer cells. Or perhaps less aggressive versus more aggressive cancer cells. The siRNA experiment in 1B-C does not prove that it is the transcript (and not the protein) that drives the phenotype.

We agree with our Reviewer on the critical issue of the reliability of in vitro and in vivo models. We always pay close attention and devote great effort to generate, test, and study our models. Nevertheless, "models" remain proxies of the physiological conditions, and the results obtained must be carefully weighed.

As characterized in the literature both through our contribution (Alaimo et al., 2020 CDDis; Genovesi et al., 2022 Biomolecules; Lunardi et al., 2021 Pathologica; Alaimo et al 2024 in preparation) and that of other groups (Tsavaler et al., 2001 Cancer Res; Henshall et al., 2003 Cancer Res; Fuessel et al., Int J On col 2003; Bidaux et al., J Clin Invest 2007; Naziroglu et al., Redox Biol 2018)), TRPM8 RNA and protein expression in both normal and malignant prostate tissue is highly heterogeneous (Figure RR1A, B from Alaimo et al., 2020), although levels almost invariably increase in PCa compared with the corresponding normal tissue (Figure RR1C, D from Alaimo et al., 2020 CDDis). Nevertheless, a new panel of expression data from three sets of samples consisting of matched primary PCa and normal prostate tissue (these data are included in a newer manuscript in preparation) confirm the high level of heterogeneity among patients with 4- to 20-fold changes of TRPM8 RNA expression between healthy and diseased tissue (since TRPM8 expression is highly specific for normal and malignant prostate luminal cells these data can be affected by the proportion of non-luminal cells in each sample). Panels E and F of Figure EV2 show increased nuclear accumulation of NF- κ B in cells expressing 50-fold more of the noncoding mutant form of TRPM8 (MM) RNA compared to the endogenous transcript, which in our opinion is a good approximation of real conditions.*

Figure RR1. TRPM8 RNA and protein expression in normal prostate tissue and prostate cancer

Leaving aside the data obtained with exogenous TRPM8 RNA, which proved crucial to untangle the coding-independent role of the transcript, the critical function of the endogenous TRPM8 RNA in TLR-3/NF- κ B activation is well-demonstrated along the manuscript. The effect of TRPM8 knockdown on PSA (Figure 1B, C) is unlikely to be dependent on reduced levels of TRPM8 protein since both a potent agonist (WS-12) and a potent antagonist (AMTB) of the channel do not elicit any effects on PSA levels.

Importantly, EVs collected from LNCaP wild type cells, but not from TRPM8-KO LNCaP (CAS), can stimulate TLR-3/NF- κ B pathway in TRPM8-KO LNCaP (CAS) cells (Figure 3E). Although the presence of TRPM8 protein in EVs cannot be excluded, we have shown that EVs certainly carry endogenous TRPM8 RNA (large fragments of hundreds of nucleotides) (Figure 3 A, B and EV3A) and that some of these fragments directly bind to TLR-3 in prostate cells (Figure 3 H-J and new Fig EV3F). Since TLR-3 is indisputably a dsRNA sensor and its pharmacological inhibition blocks the downstream signal, we can reasonably conclude that secretion of the endogenous TRPM8 transcript by wild type LNCaP cells can stimulate a sterile inflammatory condition in the same cells via autocrine and paracrine signaling.

2) Their response to R2 Q3 - Unclear what they are referring to - wrong references - New 3E? Where is the data for RWPE-1 (+/- EV)?

We apologize for the confusion, the answer to the reviewer's question is shown in Figure 2G and H and not Figure 1G and H as erroneously reported in the previous rebuttal. The experiment aimed to compare purified EVs versus EV-free supernatant was performed in LNCaP cells only for the practical reason to have the possibility to compare the efficacy of EVs and EV-free supernatants collected from CAS, WT and MM cells. RWPE-1 wild type cells express too low levels of the endogenous TRPM8 transcript, as extensively described along the text.

3) Their response to R2 Q6 - This experiment does not answer the question. The question is that conditioned media might contain a factor besides TRPM8 containing EVs that drive this process. One way to address this question would be to use EV deplete conditioned media.

We agree with our Reviewer that Figure EV5 I, J demonstrates that CAFs respond to the LNCaP MM supernatant via TLR3, which does not automatically imply the involvement of EVs. The connection between TRPM8 RNA and the altered characteristics of NAFs and CAFs in the prostate is of absolute interest for our future studies and will require dedicated efforts to be unraveled from a molecular point of view. We have toned down our statements in the revised text, as shown below by the sentence highlighted in yellow:

“Normal Associated Fibroblasts (NAFs) and Cancer Associated Fibroblasts (CAFs) were either maintained in their medium (CTR) or conditioned with the supernatants of LNCaP WT, CAS, and MM. Western blot analysis showed a substantial reduction of COL1A1 protein only in CAFs exposed to the MM supernatant (**Fig 4F**). The TCGA datasets underscored a mild but significant anticorrelation between *TRPM8* and *P3H3* and *COLGALT1*, two essential genes of the metabolism of type I collagen (Fig EV5G). Both genes resulted significantly downregulated in CAFs conditioned with the supernatant of LNCaP MM cells (**Fig 4G**; Fig EV5H), whereas their expression did not change in CAFs treated simultaneously with TLR3 inhibitor (Fig EV5I and J), which seems to suggest, but not prove, a possible role of EV-transported TRPM8 RNA in shaping the functions of CAFs.”

4) Their response to R2 Q6 - This response is incorrect and not consistent with the current use of PSA in the clinic. PSA responses (in the blood) to androgen deprivation therapy are nearly universal. Furthermore, the results remain puzzling.

We apologize with our Reviewer if the answer was unclear. Circulating PSA is the ‘universal’ biochemical marker to track therapy (not only hormone therapy) efficacy in prostate cancer patients. In almost the totality of cases PSA levels drops after therapy but the percentage of reduction as well as the nadir PSA (the lowest level to which PSA declines following treatment) are highly variable and may have prognostic value, as for example accurately described by Maha Hussain and colleagues in *The Journal of Urologists* this year (<https://doi.org/10.1097/JU.0000000000003084>).

From Hussain paper:

Purpose:

“This post hoc analysis of PROSPER evaluated the relationship between depth of PSA decline and clinical outcomes in enzalutamide-treated men with nonmetastatic castration-resistant prostate cancer”

Conclusions:

“There was a statistically significant correlation between greater depth of PSA decline and improved clinical outcomes, suggesting a previously underappreciated relationship between changes in PSA levels and clinical outcomes in nonmetastatic castration-resistant prostate cancer”.

Another important aspect that is sparking interest in the clinic is the time to nadir PSA. In the Results section, Hussain and colleagues write “In total, 65% of enzalutamide-treated men achieved a nadir PSA of $\geq 90\%$ decline from baseline, and median time to nadir PSA (interquartile range) was 227 days (116-449)”. In our opinion this represents a quite different PSA kinetics, which, in turn, likely suggests different tumor responses.

A second example that may help better clarify our views is the work recently presented at the ASCO 2022 by Andrew J Armstrong and colleagues who reported radiographic progression in the absence of PSA progression in patients with metastatic hormone-sensitive PCa enrolled in the ARCHES trial (DOI: 10.1200/JCO.2022.40.16_suppl.5072). Here their conclusion: “In this post hoc analysis of ARCHES, we found frequent discordance between radiographic progression and PSA progression by PCWG2 criteria or any PSA rise over nadir in patients with mHSPC treated with ENZA+ADT. Thus, regular imaging is recommended to detect radiographic progression among patients treated with potent androgen receptor pathway inhibitors, such as ENZA+ADT, as serial PSA monitoring alone may not be sufficient to detect radiographic progression in many patients.”

**UNIVERSITÀ
DI TRENTO**

**Department of
Cellular, Computational and Integrative Biology - CIBIO**

Overall, a growing body of clinical and preclinical evidence is revealing more and more aspects related to the biochemistry of PSA, delineating previously underestimated assets and liabilities of this gene/protein. In this scenario, we consider the identification of precise molecular pathways controlling KLK3 gene expression independently of androgens or influencing the diffusion of PSA in the blood stream important findings that may help refine and hopefully improve the clinical relevance of this crucial marker of treatment efficacy in prostate cancer.

To make our statement clearer, we have rephrased the Discussion as follows:

By promoting androgen-independent transcription of the KLK3 gene via NF- κ B and intratumor sequestration of PSA shaping the tumor microenvironment, sterile inflammation induced by PCa cells secretion of TRPM8 RNA in EVs may introduce a further layer of complexity in the evaluation of PSA levels or PSA derivatives affecting our ability to monitor the disease (PSA doubling time, PSA velocity, biochemical recurrence, biochemical response/progression).

Dear Dr Lunardi,

Thank you for submitting the revised version of your manuscript. I have now evaluated your amended manuscript and concluded that the remaining minor concerns have been sufficiently addressed.

Thus, I am pleased to inform you that your manuscript has been accepted for publication in the EMBO Journal.

Please note that it is The EMBO Journal policy for the transcript of the editorial process (containing referee reports and your response letter) to be published as an online supplement to each paper. I would accordingly like to ask your consent on keeping the referee figures included in this file.

If you do NOT want the transparent process file published, you will need to inform the Editorial Office via email immediately. More information is available here: https://www.embopress.org/transparent-process#Review_Process

On a different note, I would like to alert you that EMBO Press offers a format for a video-synopsis of work published with us, which essentially is a short, author-generated film explaining the core findings in hand drawings, and, as we believe, can be very useful to increase visibility of the work. This has proven to offer a nice opportunity for exposure i.p. for the first author(s) of the study. Please see the following link for representative examples and their integration into the article web page:

<https://www.embopress.org/doi/full/10.15252/emj.2019103932>

If you have any questions, please do not hesitate to call or email the Editorial Office.

Best regards,

Daniel Klimmeck

Daniel Klimmeck, PhD
Senior Editor
The EMBO Journal
EMBO
Postfach 1022-40
Meyerhofstrasse 1
D-69117 Heidelberg
contact@embojournal.org
Submit at: <http://emboj.msubmit.net>
